# Consensus and Subjectivity of Skin Tone Annotation for ML Fairness

**Candice Schumann**
Google
United States
cschumann@google.com

**Gbolahan O. Olanubi**
Google
United States
femio@google.com

**Auriel Wright**
Google
United States
aurielwright@google.com

**Ellis Monk, Jr.**
Harvard University*
United States
emonk@fas.harvard.edu

**Courtney Heldreth**
Google
United States
cheldreth@google.com

**Susanna Ricco**
Google
United States
ricco@google.com

## Abstract

Understanding different human attributes and how they affect model behavior may become a standard need for all model creation and usage, from traditional computer vision tasks to the newest multimodal generative AI systems. In computer vision specifically, we have relied on datasets augmented with perceived attribute signals (*e.g.*, gender presentation, skin tone, and age) and benchmarks enabled by these datasets. Typically labels for these tasks come from human annotators. However, annotating attribute signals, especially skin tone, is a difficult and subjective task. Perceived skin tone is affected by technical factors, like lighting conditions, and social factors that shape an annotator's lived experience.

This paper examines the subjectivity of skin tone annotation through a series of annotation experiments using the Monk Skin Tone (MST) scale [59], a small pool of professional photographers, and a much larger pool of trained crowdsourced annotators. Along with this study we release the Monk Skin Tone Examples (MST-E) dataset, containing 1515 images and 31 videos spread across the full MST scale. MST-E is designed to help train human annotators to annotate MST effectively. Our study shows that annotators can reliably annotate skin tone in a way that aligns with an expert in the MST scale, even under challenging environmental conditions. We also find evidence that annotators from different geographic regions rely on different mental models of MST categories resulting in annotations that systematically vary across regions. Given this, we advise practitioners to use a diverse set of annotators and a higher replication count for each image when annotating skin tone for fairness research.

## 1 Introduction

Machine learning models are ubiquitous in today's society, influencing almost every aspect of our lives. Widespread adoption brings with it heightened scrutiny, with a focus on responsible model creation — from data curation to evaluations — evident in regulation and policy discussions [24, 40, 67]. As a result, understanding different human attributes and how they affect model behavior may become a standard need for all model creation and usage, from traditional computer vision tasks to the newest multimodal generative AI systems [41]. In computer vision specifically, we foresee skin tone becoming a significant part of this conversation. Indeed, the past few years have seen

---

*Ellis Monk performed this work while also a Visiting Researcher at Google.

37th Conference on Neural Information Processing Systems (NeurIPS 2023) Track on Datasets and Benchmarks.

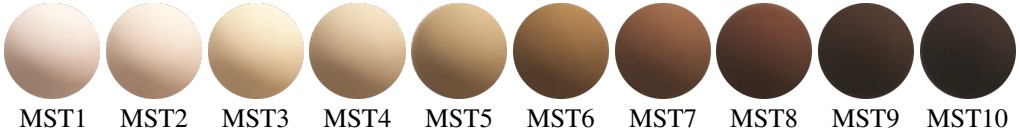

Figure 1: Monk Skin Tone (MST) scale [59].

significant investment toward computer vision models that work well for everyone, enabled by improved standards for evaluation [8, 71], better data curation and analysis [22, 81], and algorithmic advances [4, 64, 76]. To leverage these advances, many practitioners rely on image datasets enriched with third-party fairness annotations, such as perceived age, gender presentation, and skin tone [77, 82]. In this paper we focus on the collection of fine-grained perceived skin tone annotations via third party annotators.

Our focus on skin tone is inspired by research in the social sciences and machine learning. Research shows that perceived skin tone affects how we view and treat other people. Specifically, it affects interpersonal interactions [44, 62, 65], job applications [12], interpersonal relationships [17], wages, educational attainment and occupation, and health outcomes among minority ethnic and racial groups in the United States [31, 52, 54, 55, 60, 61, 63, 74].

Machine learning algorithms often perform poorly on images of people with darker skin tones [38] reinforcing the importance of considering these subgroups during model selection. Skin tone measures have been a valuable tool allowing fairness researchers to address top-of-mind challenges [16]. Researchers can devise metrics to measure diversity and representation in dataset collection [70], image generation [5, 14], and image retrieval systems [10, 58]. Finer-grained skin tone scales, such as MST (see Figure 1), promise to further improve the accuracy and fairness of machine learning systems.

Annotating skin tone is an important but subjective task [34, 35, 48]. These subjective judgements can be affected by social factors and lived experience [18, 72]. For example, social factors such as political affiliation [9], or the inferred race/ethnicity of an individual [30] can affect the perception of skin tone. Noisy estimates can lead to incorrect causal attribution of model bias [42], and over-indexing on outlier annotations or using annotations that are unduly affected by transient environmental factors can overestimate measures of diversity and representation. We must address these challenges to ensure that downstream analyses using skin tone annotations are trustworthy.

This raises the question: What is the best method to systematically annotate skin tone to build datasets and benchmarks for fairness research? In this paper we focus on the following major questions:

a) Who can annotate skin tone and what are the implications for different types of annotators?

b) Does annotator location, a proxy for cultural context and lived experience, matter and have any effect on annotation quality? If so, how can this best be managed?

c) How can practitioners best navigate the complexity of skin tone annotation and are there any best practices to consider?

To address these questions, we release the MST-E dataset, a curated dataset of exemplar images with MST annotations, to help train new annotators and study the behavior of annotator pools. Based on the results, we provide suggestions for best practices for designing skin tone annotation tasks.

## 2 Background and Related Work

In this section we look into related work on subjective annotations, skin tone, and fairness in computer vision. For a more in-depth look at work in these areas see Appendix A.

**Subjective Annotations**    Soliciting annotations for subjective concepts has been a subject of considerable research across machine learning, from natural language processing (NLP) and affective computing to classical computer vision tasks such as attribute recognition. Goyal et al. [32] found differences in annotations of toxicity in online conversations when using specialized annotator pools comprised of individuals identifying as African American and LGBTQ, respectively. Aroyo et al. [2]

annotate conversations for safety using annotator pools across two geographic regions. In both papers the authors significantly increased replications compared to standard practice, finding differences in annotator behavior between groups where a single annotator's judgements would not necessarily be stable across multiple views of the same content. Díaz et al. [20] argue for careful design of annotation tasks, accounting for the socio-cultural backgrounds of annotators and considering lived experiences as aspects of expertise. Given annotations for a subjective task from a diverse set of annotators, researchers are increasingly adopting model architectures that target individual annotators instead of or in addition to an overall consensus estimate [15, 18, 37]. Our work adds to the growing literature that considers the importance of better understanding and modeling the individual perspectives of annotators for subjective tasks, and extends the general recommendations for practitioners by focusing specifically on skin tone annotations for fairness in computer vision.

**Skin Tone Research** The Fitzpatrick Skin Type (FST) scale has become a gold standard of subjective skin tone measurement in dermatology [26]. The six-point scale, initially conceived as four-points, was developed as a dermatological tool to measure the response of different types of skin to phototherapy, exposure to ultraviolet light. Despite its wide use, there are some notable limitations of FST for skin tone annotation. In particular, the scale was not developed to assess skin tone; rather, skin, hair, eyes, and history of sunburn were used to assess likelihood of skin burning during phototherapy. In addition, to make the scale more inclusive of people with Asian, Indian, and African heritage, Type V and Type VI were later added to the four-point scale [27]. Even with the two darker additions, FST has been criticised for being skewed toward lighter skin tones [23, 29, 78] with researchers calling for finer grain skin tone measurements within dermatology [66] and for fairness in computer vision [3, 59]. That being said, fine-grained measurement is not without its own challenges and limitations.

Skin tone scales with too many shades, like the 36-point Felix von Luschan Skin Color chart [80], may be too granular and introduce additional cognitive load for annotators during an already challenging task, potentially reducing inter-rater agreement. In addition, it may be difficult for annotators to identify differences between skin tone shades at such fine-grained measurement, leading to lower annotator accuracy compared to an expert ground truth assessment.

We have chosen to have annotators use the 10-point MST scale (Figure 1). At 10 points, the MST scale was designed for a broader and more diverse range of skin tones compared to the FST without having too many options and overburdening annotators [59]. Although a finer-grained scale adds more complexity and difficulty to achieve consensus at the annotation stage (compared to, for example, annotations for a binarized skin tone annotation task), the MST allows for a more nuanced understanding of system behavior, better targeted disaggregated evaluations and audits, and more capable measures of representation. In addition, over the past year the MST has been adopted by computer vision fairness researchers to expand fairness annotations and evaluate models [14, 69]. We focus on exploring annotator behavior through the MST scale to better understand its potential as a more inclusive skin tone measurement tool in computer vision.

**Annotations for ML Fairness in Computer Vision** Perceived signals such as gender expression, skin tone, and age are commonly used in fairness evaluations and bias mitigation techniques in computer vision [8, 77, 82, 84]. There are four main methods currently used in collecting these annotations: third party crowdsourced annotations, annotations from captions, self reported annotations, and algorithmic or model generated annotations.

The most scalable solution for obtaining fairness annotations is through automated annotations with models or objective measures. However, most fairness annotation models require face or person detection or other preprocessing steps, and variable performance of upstream components across skin tone groups can lead to missing annotations for underrepresented groups during fairness evaluations [77]. In addition, these models must first be trained on annotated data. Alternatively, prior work has used measurements computed from pixel values of detected skin regions such as ITA [11] or matched skin tone pixels directly to an RGB scale [14]. In both cases these results can vary due to the influence of scene lighting or other imaging artifacts, making them less reliable as an annotation tool for in-the-wild datasets [25, 42, 43, 45, 48] (see Appendix G).

Image captions are often used to provide gender fairness annotations for images [57, 68, 86, 85]. However, relying on captions for skin tone fairness indicators is not a viable option because captions in existing datasets rarely include description of an image subject's skin tone.

Hazirbas et al. [38] released the Casual Conversations Dataset which includes both self reported gender and age. Datasets with self-reported annotations are expensive to collect and tend to be very "clean" and not in-the-wild images or videos. There is also an important distinction between subjective attributes such as skin tone, non-subjective attributes such as gender identity and age, and the perception of these attributes. While self-reported values are extremely useful and important [38], the choice to use self-reported values vs third-party annotated values depends on the perspective practitioners want to understand: a) how models interact with the identities and non-subjective attributes of users or b) how models perform on different *perceived* attributes across a wide range of imagery (including generated imagery). We note that both Hazirbas et al. [38] and the expansion in Porgali et al. [69] choose the perspective of third-party annotations for skin tone.

Data collected with annotations from topical experts and crowdsource annotators are common in computer vision. For skin tone, these annotations are often based on FST using dermatologists as annotators [8, 38, 46] due to their expertise with the FST as a dermatological assessment tool. However, topical expert annotation is not scalable to larger datasets, and has been reserved for relatively smaller annotation exercises. Crowdsourced annotations leveraging non-expert annotators has been a scalable alternative to expert annotation [38, 85]. Critically for this work, studies find that expert and crowdsource annotators tend toward fairly similar annotations on the FST [33, 48, 50]. In this paper we take a third party annotation approach and use both topical expert and crowdsource annotators using the MST scale.

## 3    The Monk Skin Tone (MST) Scale and the MST-E Dataset

The Monk Skin Tone (MST) scale defines 10 representative categories using exemplar color swatches, either as flat patches, or spheres which also incorporate shading into the stimuli. Figure 1 shows these spheres. An annotator's task is to translate from this "illustrative reference" [59] to assess the skin tone of people depicted in images captured under varying environmental conditions.

This paper introduces the open-access MST-E dataset[2] consisting of images from 19 subjects spanning the 10 point scale (see Figure 2). The goal of this dataset is to enable and test consistent skin tone annotation across environment capture conditions. It provides a starting point to support practitioners to teach their annotators to annotate skin tone consistently on the fine-grained MST scale and a controlled set through which to study annotator behavior. See Appendix C for the MST-E data card.

MST-E models, and subsequent images, were sourced through TONL[3]. We selected 30 candidate subjects from an initial pool of hundreds of models, 19 of whom were eventually photographed The 19 subjects span the full MST scale and include individuals of different ethnicities and gender identities to help human annotators decouple the concept of skin tone from race.

All images and videos of a subject were collected within a short timespan of 4 hours to eliminate skin tone variation across images due to seasonal or other temporal effects. Each subject was photographed in various poses, facial expressions (smiling, neutral, sad/frown face, and silly face), and lighting conditions. Some subjects provided a video of themselves talking to the camera, and some subjects provided images of themselves with a facial mask. See Figure 2 for example images of each of the 19 subjects. In total, the MST-E dataset contains 1515 images and 31 videos across all subjects.

The images in the dataset were reviewed and annotated using the MST scale by the authors as well as the MST creator, Dr. Ellis Monk, an expert in social perception and inequality. Images were first annotated by an author and sent to Dr. Monk for final golden annotations. Dr. Monk selected an image for each subject that best represents their skin tone (called the Golden Images set) and approved the corresponding MST annotation. We refer to these annotations as the *oracle* or *intended* annotations for the images. We curated a corresponding Challenging Images set which mirrored the Golden Images set with different lighting conditions. See Figure 4(b) for an example of "golden" versus "challenge" images and Appendix B for the full Golden Images and Challenging Images sets.

**Consent and terms of use**    Each subject provided consent for their images and videos to be released. TONL has given permission for these images to be released and used for research, model evaluations,

---

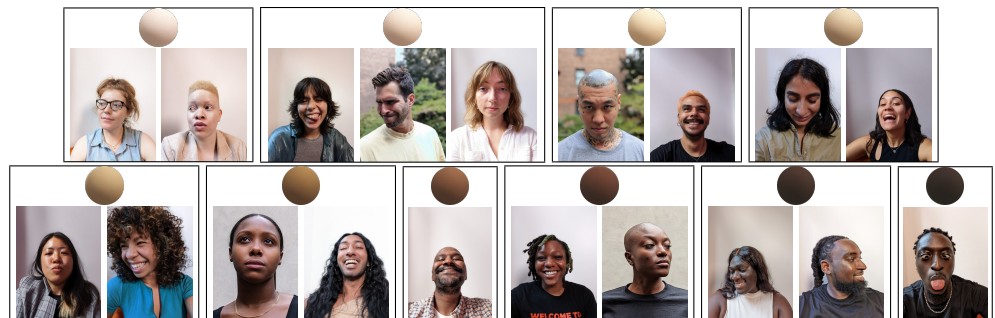

Figure 2: The MST-E dataset contains 19 subjects that span the full MST scale. Each subject has images in a variety of lighting conditions and poses. Images by TONL. Copyright TONL.CO 2022 ALL RIGHTS RESERVED. Used with permission.

or annotator-training purposes only. The images cannot be used to train machine learning models. The images and annotations are licensed under CC BY.

## 4   The Annotation Experiment

We perform two different studies in this research. The first is a qualitative survey with a group of skin tone "experts", photographers from two different geographical regions. We describe our design choices concerning the qualifications of these experts in detail in Section 5. The "experts" annotated the Golden and Challenging MST-E sets (§3). The second study expands on the results of the small scale study in two ways: extending to a larger pool of trained crowdsourced annotators located in five different geographical regions, and to an ML-training-scale dataset (§6). Crowdsourced annotators were asked to annotate skin tone across the full MST-E dataset and a subset of the Open Images dataset [49] to explore whether findings from the controlled settings generalized to practical settings. Both of these studies were conducted under the guidance of our institution's ethics body.

In each study, we examine the degree to which annotators within the same region agree with each other, whether annotators from different regions agree, and how well the consensus annotations match the intended annotation across different conditions. In the following sections, when calculating the consensus MST annotation for an image we use the median MST annotation from the responses.

Typically, the only available indicator of annotation quality is a measure of inter-rater reliability. However, in this instance we have MST annotations provided by the creator of the MST scale. Thus, we have the unique opportunity to compare annotations from the annotators to his annotations.

Previous research has taken a similar approach comparing FST annotations between crowdsource annotators and an expert dermatologist, and has found that crowdsource annotations are typically within 1-point of expert dermatologist annotations [33]. This threshold of 1-point discrepancy has continued to be used when comparing skin tone annotations between crowdsource annotators and experts [34]. Here, we consider alignment between the consensus from the photography experts or crowdsourced annotators and Dr. Monk's intent using two metrics: the percentage of median MST annotations that are within 1-point of discrepancy from the oracle annotations (higher is better), and the average median distance, the expected difference between the median annotation from a group of annotators and the oracle computed across all images (ideally less than 1).

## 5   Exploration with Experts

We used Gerson Lehrman Group (GLG) to recruit and compensate a panel of five experts in photography in India and five experts in photography in the United States to complete a skin tone annotation task. Experts in India were selected because we later recruit crowdsource annotators from this market. The U.S. was selected because we sought a geographically distant comparison market in which GLG could recruit a similarly sized panel of experts. Similar skin tone annotation research has sought annotations from dermatologists [34], but we did not want experts who were practiced in reviewing images taken under highly controlled conditions for medical issues. Rather, we wanted experts who

were practiced in capturing and editing still images of people and examining skin in their photography. During an initial pilot we found that different types of experts may a) approach the task differently based on their discipline and b) this might result in differences in annotation. Specifically we found that photographers tended to approach the task as designed, assessing the skin tone in the image while correcting for extraneous image features. However, dermatologists interviewed in the pilot study were more likely to attempt to assess skin tone beneath clothing which was not depicted. This aligned with skin tone assessment in a medical scenario, but not with our goal of achieving skin tone annotation in images that aligns with *social perception*. For this purpose we recruited photographers who self-reported that people and portraits were the subjects of their work, had previous experience reviewing and editing skin tone in color correction software, and were most likely to approach the task as designed.

Experts were asked to annotate a total of 50 images, 19 from the MST-E Golden Images or Challenging Image sets (§3) and 31 aesthetically similar distractor images. Images were presented in a randomly assigned order and the participants were asked to view each image and select which skin tone on the MST best matched the person depicted in the image. The task was completed twice. Participants first annotated the Golden Images set and distractor images. Approximately one week later, participants annotated the Challenging Images set and a new set of distractor images.[4] The experimental setup was designed to examine consistency of annotation across lighting conditions, with the time delay reducing the likelihood that participants would annotate the Challenging set based on their memory of their previous selection.

Experts were shown an image and asked to select the skin-tone that matches most closely on the MST scale, to select which part of the person's body they used in their assessment, and to rate on a 5-point scale the ease/difficulty of the annotation. An example of the task can be found in Appendix D.

**Inter-Rater Reliability**  We examine if the skin tone annotations are consistent among experts within each market using intraclass correlation (ICC), a measure of inter-rater reliability. Each expert reviewed each image, so we use a two-way random, single score ICC where the subject and annotator are viewed as random effect [56]. We find that experts in the tested markets do consistently annotate skin tone on the MST. Figure 3(a) shows significant, p <.05, intraclass correlation in both markets and in both image quality conditions.

**Regional Differences in Annotation**  We conducted a series of Mann Whitney U tests [53] on each individual image to determine if there were any significant differences in the annotation between the experts from the United States and those in India (see Appendix E for full comparisons). Differences were significant using $p < 0.05$ for three of the images. Figure 4 shows the annotations for two of these in detail. The professional photographers in India tended to annotate the subject's skin as lighter (MST3 - MST5). Professional photographers in the U.S. tended to annotate the subject's skin as darker (MST5 - MST7). The differences observed provide suggestive evidence that geographic location affects skin tone annotation, which we explore further and at scale using crowdsourced annotators.

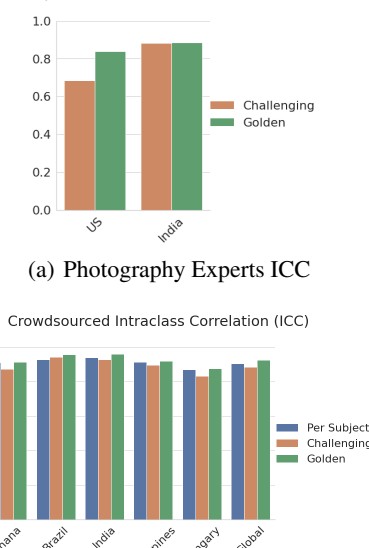

(a) Photography Experts ICC

(b) Crowdsource ICC

Figure 3:  Inter-rater reliability (IRR) between annotators calculated via intraclass correlation (ICC). Figure 3(a) shows the ICC scores for expert photographers in the US and India on the Challenging set and the Golden set. Figure 3(b) shows the ICC scores for crowdsourced annotators in each region, as well as the global pool. These ICC scores are calculated per subject, as well as over the Challenging and Golden sets.

---

[4]One of the MST-E subjects was excluded from this study due to a programming error.

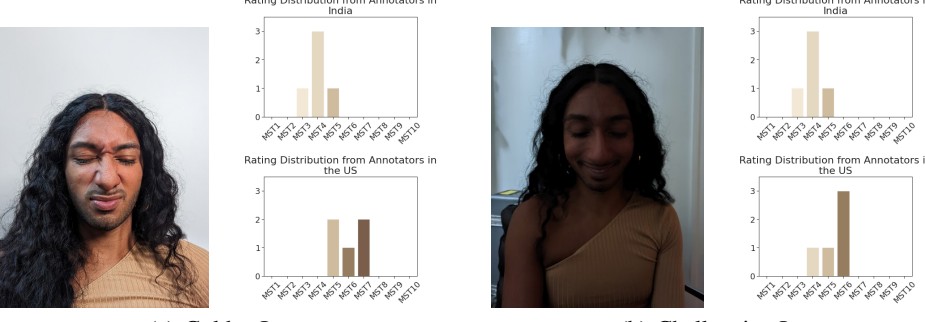

(a) Golden Image          (b) Challenging Image

Figure 4: Photography experts in India (top plots) and the US (bottom plots) MST annotation counts for Subject 14. Figure 4(a) shows Subject 14's Golden Image chosen by the MST scale expert (§3). Figure 4(b) shows Subject 14's matching Challenging Image. In large-scale experiments with trained annotators (§6), we found consistently measurable differences in annotator behavior based on region.

**Distance from Intended Annotations**  In both markets we found that a high percentage of the consensus annotations were with 1-point of discrepancy of the oracle annotations, 88.9% in India and 83.4% in the United States. In addition, in both markets the average median distance from the oracle annotation was not above the 1-point guardrail established in previous research [33, 34], 0.78 in India and 0.84 in the United States. These results suggest that photography experts in India and the U.S. annotate skin tone in a manner that aligns with the scale creator's intent.

## 6  Exploration of Crowdsourced Annotations

We collected annotations for the entire MST-E dataset and a subset of the Open Images dataset from annotators in India, the Philippines, Brazil, Hungary, and Ghana. We followed standard processes for data annotation for computer vision-related work, using the services of professional data annotators without specific domain expertise outside of the instructions they received for this task. Annotators are compensated for their work (see Appendix K for more details). Each task was replicated 5 times in each region resulting in 25 annotations per image.

During a task, annotators are shown a thumbnail of a subject and the full image with a box drawn around the location of the thumbnail. This allows annotators to understand the context and lighting of the image. Annotators were asked to select the best skin tone for the subject and report which part of the body they primarily used to make the selection (Face, Torso, Arms/Hands, Legs/Feet, or Other). Annotators could select that an image was difficult to annotate for skin tone, and skip the task if no skin was visible. See Appendix D for an example of the selection tool.

Since the MST-E dataset contains many images for each subject, the images were mixed into a much larger set of similar images to prevent face recall among annotators. In the results section we look at the annotations *Per Subject* - meaning annotations across all images of a single subject - as well as the annotations for the Golden Images and Challenging Images sets (§3).

**Inter-Rater Reliability**  Five annotators were randomly selected from a broader pool of annotators within each market to annotate each image. As the group of annotators was not consistent from image to image, we use a one-way random, single score ICC to measure inter-rater reliability, where the subject is a random effect for this analysis [56].

Consistent with our results from professional photographers, we find that non-expert annotators also consistently annotate skin tone on the MST, using a shared mental model of the MST categories. Figure 3(b) shows significant, $p < 0.05$, ICC values for each region and annotation task. For every region, the ICC is very high with Per Subject ICC ranging from $0.86$ to $0.94$. In addition, we find some evidence that annotators remain consistent in skin tone annotation even when image quality is diminished. The Golden Images ICC ranged from $0.9$ to $0.96$, and Challenging Images ICC ranged from $0.82$ to $0.95$ – a comparatively lower range but still significant and consistent.

**Regional Differences in Annotation**    We found suggestive evidence for regional differences in annotation in Section 5, and know that geographical region affects annotations in other domains like opinion of the safety of chatbots [2]. Unlike the exploratory analysis of experts that used a subset of the MST-E dataset, in this setting we have enough statistical power to explicitly test the effect of the annotator's region on MST annotation. We performed a multivariate analysis of variance (MANOVA) [28] with annotator region as the independent variable and MST annotations for each subject as the dependent variable. We found that the subject annotations were a significantly high predictor of the region of the annotators (Pillai's trace = 1.97, $p < 0.05$). This result suggests that the shared mental models of skin tone may vary significantly across different geographic regions.

**Distance from Intended Annotations**    Despite measurable regional variation, we find a high percentage of median MST annotations that are within 1 point of oracle annotations. Using the global pool, the consensus annotation agreed within 1 point for all but one subject (94.7%). In addition, we find the average median distance to oracle annotations to be less than 1 for every region and image condition. The distance was 0.78 for both well lit and poorly lit images. This suggests that crowdsource annotators, similar to the expert photographers, arrive at consensus MST annotations that are similar to the scale creator's intent.

**Replications**    The 25 annotations we acquired per image is larger than most practitioners collect. We use the Spearman-Brown prediction formula to determine inter-rater agreement over $k$ replications, where $k$ is the total number of replications over all regions (See Figure 5) [7, 79, 83]. We find that for datasets containing high quality images or annotations of a single individual based on a consensus across multiple images, one rater per region likely suffices (5 total annotators). For more challenging images, two annotators per region is beneficial. Therefore, if a practitioner's budget is limited, we would suggest filtering for images with good lighting conditions (or even images with okay lighting conditions). If annotations on images with low quality lighting is necessary, practitioners can then annotate with 10 annotators on just those low quality images.

**Extension to In-the-Wild Annotations**    Open Images is a large-scale image dataset with a number of image and object annotations [49]. We randomly selected a subset of 41,471 images that contained at least one face. In the 41,471 images, 86,809 faces were present to annotate. The annotators used the same annotation tool as they did with the MST-E dataset (see Appendix D). Using this tool annotators could see the face thumbnail along with its location in the full image.

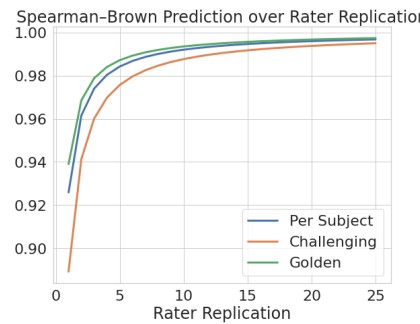

Figure 5: Spearman-Brown inter-rater reliability calculations over replications of annotators on the MST-E dataset per subject, over the Challenging Images, and over the Golden Images.

Unlike the MST-E dataset where images were taken in a controlled setting, Open Images is an "in-the-wild" dataset containing many more unique individuals and images of more varying quality. Even in these uncontrolled settings we expect that crowdsource annotators will still consistently annotate skin tone on the Monk Skin Tone scale and that geographic region will have a significant effect on the annotations.

Consistent with our hypotheses, we find a high intraclass correlation across markets ranging from 0.875 to 0.91. In addition, the intraclass correlation from the Open Images dataset has no significant differences than crowdsource annotators intraclass correlation in the MST-E dataset ($\chi^2 = 0.26$, $p = 0.99 > 0.05$). This suggests that crowdsource annotators are similarly consistent in MST annotations in controlled datasets and in-the-wild data sets. We also find significant differences between annotation distributions between regions ($V^2 = 0.0766$, $p < 0.05$), consistent with the MST-E annotations. Overall, this shows that recommendations based on annotator behaviors observed in our controlled studies likely translate to real-world scenarios commonly faced by ML practitioners.

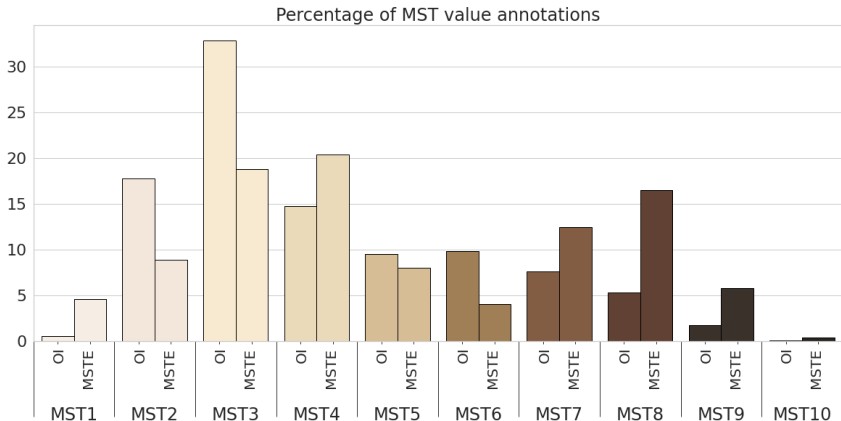

Figure 6: Comparison of the distribution of skin tone annotations in the MST-E dataset ("MSTE" in the plot) versus the "in-the-wild" Open Images dataset ("OI" in the plot). The Open Images dataset has a strong skew towards lighter skin tones.

# 7    Discussion and Suggestions for Practitioners

One of the first design decisions practitioners have to make when starting a skin tone annotation task is the composition of the annotator pool. Following from common practices in literature [8, 85], we analyse MST annotations from subject matter experts, professional photographers who focus on photographing people, and crowdsourced annotators who were provided training in MST skin tone annotations prior to the annotation task. We find that both the photographers and trained annotators were able to reliably annotate for skin tone suggesting that subject matter experts are not necessarily needed for MST skin tone annotation tasks. This allows for more scalable annotation tasks using trained annotators on large datasets.

Additionally we find that although trained annotators have high inter-rater reliability across all regions, there are regional differences between the MST annotations, suggesting that regional cultural contexts influence how people perceive skin tone. When combining the annotations across a variety of regions, we find that the median of the annotations are consistently close to Dr. Monk's annotations. This suggests that a geographically diverse annotator pool provides annotations that are in line with the intended use of the MST scale [59].

We find that annotators are able to learn to be invariant to environmental factors such as lighting quality, and find that they perform similarly on the in-the-wild subset of Open Images to the MST-E dataset, suggesting that performing a skin tone annotation task on in-the-wild images is possible.

The MST-E dataset is a valuable tool to help practitioners train their human annotators to annotate MST on images. But beyond that use case, the dataset can also be used to evaluate models. Particularly, the MST-E dataset can be used to evaluate fairness over the full MST scale as well as to evaluate performance of models over varying lighting conditions and poses. This is particularly useful since "in-the-wild" datasets like Open Images may have a large skew in skin tone distribution (See Figure 6).

**Limitations and Ethical Considerations**    This paper discusses the variance introduced by different geographical regions and types of annotators. However, it does not consider other aspects of annotator identity and background, or other factors such as the language used in the task, different UI improvements, and if annotators are affected by other people in the same image. Future work should look at these issues.

This work suggests the use of 10 annotators per image if image quality is unknown. This is a very large jump from the status quo of 3 annotators per image, which could be a limiting factor for practitioners with lower budget. A potential alternative for very restricted budgets would be to use a large and geographically diverse pool of annotators but have only a subset of annotators provide explicit judgements for each image. One would then impute missing annotations based on a machine learned model personalized for each annotator. Similar approaches have been tested for other types of subjective annotations (in NLP [18] and in face similarity [1]) but not yet for skin tone. These methods should be investigated in future work.

The MST-E dataset has a small number of subjects (19), and while they do span the complete MST scale, they do not capture *every* variation in skin tone. Future work should extend this dataset to more subjects. In addition, the exploratory analyses of annotations from photography experts is more limited than the scale achieved with crowdsourced annotators. That being said, with the additional analysis of the Open Images dataset, we believe our results will translate to other domains. While we use thresholds for agreement (e.g., 1-point discrepancy) that are in line with prior work when describing the resulting annotations as reliable, future work should focus on usefulness of the annotations in fairness analysis or mitigations leading to more inclusive and robust computer vision systems.

The MST-E dataset release enables the annotation of an aspect of a person's appearance which has historically been the source of discrimination and continues to affect how people are treated in society. While there is the potential for malicious use of skin tone analysis, we believe the clear benefits for fairness research outweigh the potential risks. We chose not to collect or release other sensitive information about the subjects of the dataset or our annotators, beyond general geographic region, to protect their privacy. While we have consent from participants to release the MST-E dataset for research and annotator training purposes, there remains a risk that the images could be used maliciously in model training. In order to mitigate these risks we clearly state the terms of use and do not allow for anonymous access to the image dataset.

**Suggestions for practitioners** Extrapolating from our results, we make the following suggestions for practitioners who are pursuing a task of MST scale annotation.

- Diversity of annotators is important for subjective annotations. In our case cultural differences across geographic regions influenced skin tone annotations. Other subjective annotations may require different types of diversity (i.e., gender in heteronormative communities vs members of the LGBTQ+ community).
- Given the wide range of annotations we suggest having more than three annotators per subject (the current status quo) and instead have at least two annotators in at least five different geographical regions (10 ratings per image), particularly for challenging images.
- If you know the geographical region of your data or deployment scenario, it may be wise to use annotators from the same region. We recommend considering this design choice due to the difference in behaviors across geographic pools we observed. We hypothesize that co-located annotators may have a better cultural context for certain annotation tasks that would provide the most relevant data, although more careful study of this is needed.
- Train annotators with examples of individuals across the full MST spectrum in a variety of conditions. The MST-E dataset can be used to perform this training.

## 8 Conclusion

Fairness signals/attributes are important to determine how well models, and eventually products using these models, work for **every** person. Skin tone is particularly important for computer vision models and applications. However, annotating skin tone (even annotating one's own skin tone) is a subjective task. We make the following contributions:

a) We show that skin tone can be annotated reliably on the MST scale by both trained crowdsourced annotators and subject-matter experts such as professional photographers.
b) We show that annotators have an understanding of skin tone across conditions and, with training, can provide annotations invariant to lighting conditions.
c) We show that cultural contexts across geographical regions influence MST annotations. However, when collecting annotations from a diverse geographical pool, annotators are able to provide annotations that match the original intention behind the MST scale.
d) We provide actionable suggestions to practitioners collecting subjective annotations.
e) We release the MST-E dataset to help practitioners train annotators to be invariant to lighting conditions and support further research.

Although fine-grained skin tone annotation is a difficult task, it is possible and it opens up the door to more nuanced understanding of bias in model performance.

## Acknowledgements

We wish to thank our colleagues across Google working on fairness and inclusion in computer vision for their contributions to this work, especially Marco Andreetto, Parker Barnes, Ken Burke, Benoit Corda, Tulsee Doshi, Rachel Hornung, David Madras, Shrikanth Narayanan, Utsav Prabhu, Sagar Savla, Alex Siegman, Komal Singh and Biao Wang. We also would like to thank Annie Jean-Baptiste, Florian Koenigsberger, Marc Repnyek, Maura O'Brien, and Dominique Mungin and the rest of the team who help supervise, fund, and coordinate our data collection.

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

# A Expanded Related Work

## A.1 Subjective Annotations

Soliciting annotations for subjective concepts is required for many modern machine learning tasks. As such, it has been a subject of considerable research across subfields, from NLP and affective computing to classical computer vision tasks such as attribute recognition. Our work adds to the growing literature that raises the importance of better understanding and modeling the individual perspectives of annotators for subjective tasks, especially as influenced by demographic or cultural factors, and we highlight a few examples here from across disciplines. Goyal et al. [32] study differences in annotations of toxicity in online conversations when using specialized annotator pools comprised of individuals identifying as African American and LGBTQ, respectively. Aroyo et al. [2] annotate conversations for safety using annotator pools across two geographic regions. They significantly increased replications compared to standard practice, finding differences in annotator behavior between groups where a single annotator's judgements would not necessarily be stable across multiple views of the same content. Díaz et al. [20] argue for careful design of annotation tasks, accounting for the socio-cultural backgrounds of annotators and considering lived experiences as aspects of expertise. Given annotations for a subjective task from a diverse set of annotators, researchers are increasingly adopting model architectures that target individual annotators instead of or in addition to an overall consensus estimate [18, 15, 37]. Our work supports these findings and extends the general recommendations for practitioners by focusing specifically on how they apply to annotations for fairness in computer vision.

## A.2 Skin Tone Research

There are a number of different ways to describe skin tone that have been used in the social sciences, dermatology, and computer vision. Here, we focus on the most common definitions used in computer vision fairness research, including more options and more detail than discussed in the main paper.

The most common objective method used by computer vision scientists and AI researchers is the individual typology angle (ITA), which is a metric based on the $L*$ (lightness) and $B*$ (yellow/blue) components of the CIE $L*a*b*$ color space, defined as $ITA = \arctan\left(\frac{L*-50)}{b*}\right)\left(\frac{180}{\pi}\right)$. Skin color can then be classified using the ITA according to six categories, ranging from very light (category I) to dark (category VI) [11]. Measured ITA has been shown to be correlated with physical properties such as melanin content of skin [19]. The mathematical definition is conceptually appealing, with researchers describing it favorably as, for example, being "easily computed without subjective annotations." [43]. However, ITA values computed from images captured in uncontrolled environments can vary due to the influence of scene lighting or other imaging artifacts [73], making it less reliable as an annotation tool for in-the-wild datasets. Furthermore, in practice the continuous ITA values are often mapped to different categorical scales using boundaries between categories that reintroduce elements of subjectivity [11, 19].

The Fitzpatrick Skin Type (FST) scale has become a gold standard of subjective skin tone measurement in dermatology. The six-point scale, initially conceived as four-points, was developed as a dermatological tool to measure the response of different types of skin to phototherapy, exposure to ultraviolet light. For example Type I, the lightest skin type, is described as "Burns easily, never tans" and Type VI, the darkest skin type, is described as "Never burns, tans profusely" [27]. Outside of dermatology, computer vision scientists have also used the FST to label images for skin tone.

Most studies use either expert dermatologists or crowdsourced annotators to assign skin tone labels based on FST [46, 38]. Though lighting and image quality may pose a challenge in photo evaluation, experts and crowdsource annotators tend toward fairly accurate FST annotations where judgements are expected to be within one FST value depending on image conditions [50, 48]. For example, an audit of the Fitzpatrick 17k dataset found their crowdsourced annotators had between 71% and 85% within one point accuracy to board-certified dermatologist ratings [33].

Despite its wide use, there are some notable limitations of FST for skin tone annotation. In particular, the scale was not developed to assess skin tone; rather, skin, hair, eyes, and history of sunburn were used to assess likelihood of skin burning during phototherapy. In addition, to make the scale more inclusive of people with Asian, Indian, and African heritage, Type V and Type VI were later

added to the four-point scale [27]. These additions to the scale implied a strong correlation among race/ethnicity, skin tone, and sun reactivity; however, self-reported race/ethnicity and skin tone have been found to be significant but weak predictors of FST [39]. That is, FST values may actually serve as a significant, but weak, correlate to one's actual skin tone.

FST also better represents a range of lighter skin tones than a range of darker skin tones [27, 29]. One study found that only 42% of an ethnically diverse group of participants could be classified using the standard Fitzpatrick scale definitions [23]. Sommers (2019) [78] summarized it saying, " they [self-reported Fitzpatrick Skin Type] provide a restricted range of options for people with darker skin that does not capture variations in their skin color." This highlights a core limitation of the FST; it is not inclusive of the full spectrum of skin tones. This limits the ability for the FST to faithfully represent people, and the ability for systems built upon human annotation to be fairly evaluated across skin types.

Researchers have called for alternative and finer-grained skin tone measurements both within dermatology [66] and more broadly in fairness for computer vision [59]. The ability to draw on a finer-grained scale that reflects a broad range of skin tones provides flexibility to support careful design choices for disaggregated evaluations [3] and can play a crucial role in addressing performance disparities, especially those which disproportionately impact communities that have faced injustices and have been historically underrepresented [8]. Although finer-grained measurement is the goal, it is not without its challenges and limitations.

Skin tone scales with too many shades, like the 36-point Felix von Luschan Skin Color chart [80], can introduce additional cognitive load for annotators during an already challenging task, potentially reducing inter-rater agreement. In addition, it may be difficult for annotators to identify differences between skin tone shades at such fine-grained measurement, leading to lower annotator accuracy compared to an expert ground truth assessment.

We have chosen to have annotators use the 10-point Monk Skin Tone (MST) scale [59] (Figure 1). The MST scale was designed through its color selection and gradations to optimally capture ethnoracial diversity in skin tone across the Americas (e.g. North, South). This includes capturing skin tone variation within and across racial/ethnic categories in the United States (and potentially beyond). At 10 points, we hypothesized MST would give annotators a broader and more diverse range of skin tones compared to the FST without overwhelming annotators with too many options. Although a finer-grained scale adds more complexity and difficulty to achieve consensus at the annotation stage (compared to, for example, annotations for a binarized skin tone annotation task), we hypothesized that the MST allows for a more nuanced understanding of system behavior, better targeted disaggregated evaluations and audits, and more capable measures of representation.

### A.3 Annotations for ML Fairness in Computer Vision

Perceived signals such as gender expression, skin tone, and age are commonly used in fairness evaluations and bias mitigation techniques in computer vision [82, 8, 84]. There are four main methods currently used in collecting these annotations: third party crowdsourced annotations, annotations from captions, self reported annotations, and algorithmic or model generated annotations.

Crowdsourcing annotations for fairness evaluations is extremely common in computer vision. Zhao et al. [85] collect both Fitzpatrick Skin Type [26] and gender expression using Amazon Mechanical Turk (MTurk) with three replications per image. Amazon Mechanical Turk (AMT) is a popular crowdsourcing platform that is commonly used across both computer vision and general machine learning practitioners. Difallah et al. [21] found that the majority of MTurk workers are based in the US and India. They also found that MTurk workers tend to be younger than the overall population. Buolamwini and Gebru [8] collected Fitzpatrick Skin Type annotations for each individual through a board-certified surgical dermatologist. This type of annotation is not scalable to larger datasets. Schumann et al. [77] show how annotation pipelines can introduce biases into annotations. They also provide a dataset *Open Images Extended: More Inclusive Annotations for People* (MIAP) with crowdsourced annotations from annotators in India of perceived gender presentation and age range. In this paper we focus on crowdsourced annotations for skin tone across a wide geographic region.

Captions are often used to provide fairness annotations for images. For example, a body of work on investigating and mitigating bias in image captioning builds on the dataset and definitions introduced by Zhao et al. [86]. Here, images from MSCOCO are filtered into dataset slices depending on whether

the ground truth captions include only the word "man", or only the word "woman". Images with captions mentioning both terms are excluded. Recent work on understanding gendered bias in vision datasets [57] adopts a similar strategy to identify dataset slices, with updates introduced by Zhao et al. [85] that expand the list of gendered terms included in the filters. It is possible to construct sliced datasets of reasonable scale using this procedure because it is common for descriptions of people in the source datasets to use gendered terms. In fact, the annotation instructions in the Localized Narratives dataset includes the example "a man is flying a kite" [68], implicitly encouraging this practice.

In contrast, we find that captions in existing datasets rarely include description of an image subject's skin tone. We used *Know Your Data*[5] to find and manually inspect images containing a person bounding box and captions that contain the word *skin* or *skinned*. Out of all 123,287 images, we find just 12 images with captions describing skin tone - one "pale skin" and 11 "dark skin(ned)". Given that recent developments in large-scale vision-language models rely on paired text from web-crawled alt-text [13, 75] rather than curated captions, we consider whether alt text datasets may exhibit different properties. Hanley et al. [36] provides alt text guidelines from two museums that recommend describing skin tone in images, and notes that nearly all of the concrete examples include these descriptors. However, web-crawled alt text from generic sources is of varying quality and does not adhere to these standards [6]. Even if we could ensure museum-quality alt text that follow these published guidelines, the suggested language simplifies the concept, resorting to more binarized language like light- or dark-skinned.

Liu et al. [51] released the Casual Conversations Dataset which includes both self-reported gender and age. In contrast, the authors of the dataset use third party annotations of Fitzpatrick Skin Type, not self-reported values, to categorize the skin tone of participants. Datasets with self-reported annotations are expensive to collect and tend to be very "clean" and not in-the-wild images or videos. There is also an important distinction between subjective attributes such as skin tone, non-subjective attributes such as gender identity and age, and the perception of these attributes. While self-reported values are extremely useful and important Liu et al. [51], the choice to use self-reported values vs third-party annotated values depends on the perspective practitioners want to understand: a) how models interact with the identities and non-subjective attributes of users or b) how models perform on different *perceived* attributes across a wide range of imagery (including generated imagery).

Finally, fairness annotations can be obtained through models as explored by Buolamwini and Gebru [8]. However, these models need to be trained on annotations acquired from one or more of the above techniques. Skin tone measures based off of ITA are notable exceptions. These measures are computed from pixel values from detected skin regions in the images with the objective of matching physical measurements captured under controlled lighting conditions [19, 42, 73]. Raw pixel values in images captured under unknown lighting conditions are a function of both the scene lighting and the subject's true skin tone. Earlier work recognizes but does not always attempt to correct for these effects [43, 45]. Others apply color correction by leveraging known properties of the scene or capture environment [48, 42]. Recently, Feng et al. [25] introduce a learned method that jointly models scene illumination inferred from scene context and facial albedo, from which ITA can be computed. Cho et al. [14] present a rare instance of a method that estimates skin tone directly from pixel values but bypasses ITA computation. They detect skin regions in images and select the Monk Skin Tone category that most closely matches the average RGB value of pixels within this mask. We do not recommend this approach for MST annotations. See Appendix G for an illustration of limitations of this approach with unconstrained images.

# B  MST-E Golden and Challenging Image Sets

The MST-E dataset includes a set of *Golden Images* where each subject is represented by an image that best represents their skin tone. A *Challenging Images* set was also selected with worse lighting conditions.

---

[5]https://knowyourdata-tfds.withgoogle.com/#dataset=coco_captions

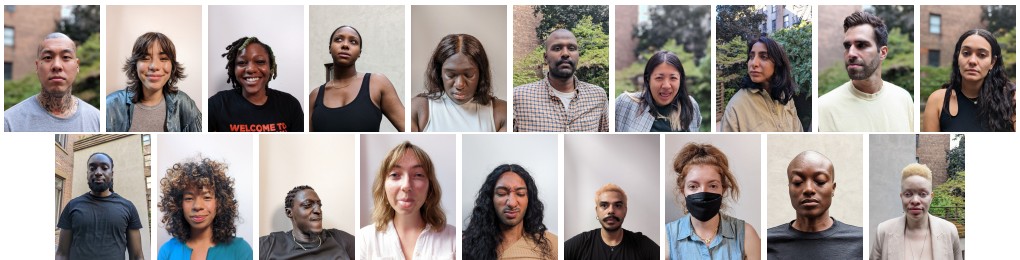

*Golden Images*

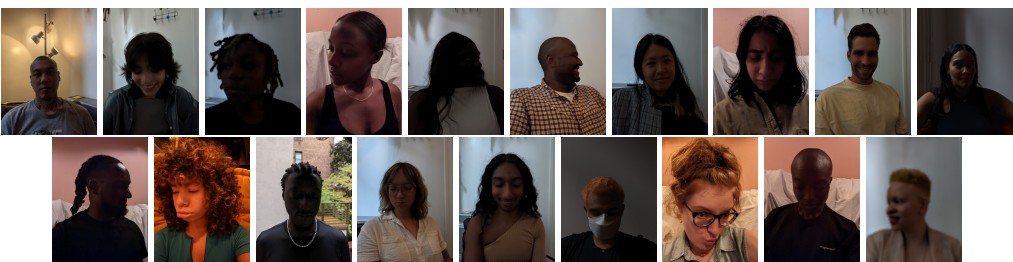

*Challenging Images*

## C Data Card

| Publishers | Team | Contact Detail |
|---|---|---|
| Google LLC
TONL.CO | Google Research, Perception and Responsible AI teams | **Contact**: Feedback form
**Website**: skintone.google |

| Data Subject(s) | Data Snapshot | Data Description |
|---|---|---|
| Data about image search queries | Dataset size 5.21 GB
Subjects 19
Images 1515
Videos 31 | The primary goal of this dataset is to enable and test consistent skin tone annotation across environment capture conditions. It provides a starting point to support practitioners to teach their annotators to annotate skin tone consistently on the fine-grained MST scale and a controlled set through which to study annotator behavior.

Images by TONL. Copyright TONL.CO 2022 ALL RIGHTS RESERVED. |

| Primary Data Modality | Link to Data | Data Fields |
|---|---|---|
| Image Data
Video Data | skintone.google/mste-dataset | Data field for each image and video can be found in a csv with the following headers (Header descriptions follow the format `Field Name`, *Example*, Description)

• `image_name`, *PXL_0.jpg*, Name of the image
• `pose`, *side*, Pose of the subject
• `lighting`, *well_lit*, Lighting condition of the image
• `mask`, *false*, Whether the subject is wearing a mask
• `subject_name`, *subject_0*, Name of the subject
• `MST`, 1, MST annotation for the subject

The Golden Image Set and Challenging Image Set can be found in a csv with the following headers:

• `subject`, *subject_0*, Name of the subject

• `golden_image_id`, *PXL_0.jpg*, Name of the golden image

• `challenge_image_id`, *PXL_1.jpg*, Name of the challenging image

The Images and Videos for each subject can be found under the subject's folder. For example, in the above examples image *PXL_0.jpg* can be found in the *subject_0* folder. |

| Secondary Data Modality | Link to Data | Data Fields Crowdsourced annotations can be found in a csv with the following headers (Header description follow the format `Field Name`, *example*, Description) |
|---|---|---|
| Image Data Crowdsource Annotations | `https://storage.googleapis.com/mste/mste_annotations.csv` | |

Data Fields Crowdsourced annotations can be found in a csv with the following headers (Header description follow the format `Field Name`, *example*, Description)

- `subject`, *subject_0*, Name of the subject
- `image`, *PXL_0.jpg*, Name of the image
- `region`, *Ghana*, Region where the annotation came from
- `annotation`, *MST1*, Crowdsourced MST annotation.

Note that each image should have 5 annotations per region with a total of 25 annotations per image. Some images were too hard to annotate and were skipped by some annotators. In those cases there will be less than 25 annotations.

| Purpose(s) | Domain(s) of application | Motivating factor(s) |
|---|---|---|
| Fairness research | `Research, Fairness, Computer Vision, Skin Tone, Model Evaluation` | <ul><li>Subjects span the entire MST scale.</li><li>Trains annotators to annotate images on the MST scale.</li><li>Trains annotators to be invariant to lighting conditions.</li><li>Supports fairness research.</li></ul> |

| Dataset Use(s) | Intended and suitable use case(s) | Unsuitable use case(s) |
|---|---|---|
| Research
Annotator training
Model Evalutation | <ul><li>**Research**: Fairness research</li><li>**Annotator training**: Can be used to train *human* annotators to annotate skin tone.</li><li>**Model Evaluation**: Can be used to evaluate fairness in computer vision models.</li></ul> | <ul><li>**Model training**: The images *cannot* be used to train machine learning models.</li></ul> |

### Version Status

**Actively Maintained**
No new versions with new images will be made available, but this dataset will be actively maintained, including but not limited to updates to the data if errors are reported.

### Dataset version

**Current Version** 1.0
**Last Updated** 05/2023
**Release Date** 05/2023

### Maintenance plan

- **Storage**: The dataset is stored on Google Cloud.
- **Versioning**: The website skintone.google/mste-dataset will contain the latest version of the dataset.
- **Availability**: The dataset will be available for download with a valid Google account. The dataset will remain available on Google Cloud.
- **Feedback**: Feedback and errors can be submitted through the Feedback form.

### Provenance | Collection | Method(s) used

Images sourced through TONL.CO.
Annotated by a human expert.

### Provenance | Collection | Methodology detail(s)

**Source**: TONL.CO and skintone.google
**Date of Collection**: 2022
**Primary modality**: Image Data
**Update frequency**: Static

### Provenance | Collection | Data Processing

**Images**: The subjects, and subsequent images, were sourced through TONL.CO.
**Timing**: All images and videos of a subject were collected within a short timespan of 4 hours to eliminate skin tone variation across images due to seasonal or other temporal effects.

### Human attributes

Skin Tone

### Source(s) of human attributes

**Skin Tone**: The images in the dataset were reviewed and annotated using the MST scale by the MST creator, Dr. Ellis Monk, an expert in social perception and inequality.

### Rationale for collecting human attributes

The primary goal of this dataset is to enable and test consistent skin tone annotation across environment capture conditions. It provides a starting point to support practitioners to teach their annotators to annotate skin tone consistently on the fine-grained MST scale and a controlled set through which to study annotator behavior.

### Distribution of human attributes

| MST | 1 | 2 | 3 | 4 | 5 | 6 | 7 | 8 | 9 | 10 |
|---|---|---|---|---|---|---|---|---|---|---|
| Subjects | 2 | 3 | 2 | 2 | 2 | 2 | 1 | 2 | 2 | 1 |
| Images | 179 | 218 | 112 | 187 | 179 | 165 | 76 | 143 | 142 | 111 |
| Videos | 2 | 5 | 5 | 3 | 4 | 3 | 4 | 2 | 5 | 1 |

| Annotation workforce type | Annotation characteristics | Annotation description |
|---|---|---|
| Human Annotations - Expert | Number of unique annotations: 19 | The images in the dataset were reviewed and annotated using the MST scale by the MST creator, Dr. Ellis Monk, an expert in social perception and inequality. Each subject was annotated with a single MST-E value. |

## Concepts and definitions referenced in this data card

### MST

Definition: Monk Skin Tone
Source: skintone.google
Interpretation: Inclusive 10 point skin tone scale.

| Annotation workforce type | Annotation characteristics | Annotation description |
|---|---|---|
| Human Annotations - Crowdsource | Number of unique annotations: | Crowdsourced annotations for the MST-E dataset has been made available for replication purposes only. Each image was annotated 5 times in 5 different regions totalling 25 annotations per image. |

## D  Annotator Experience

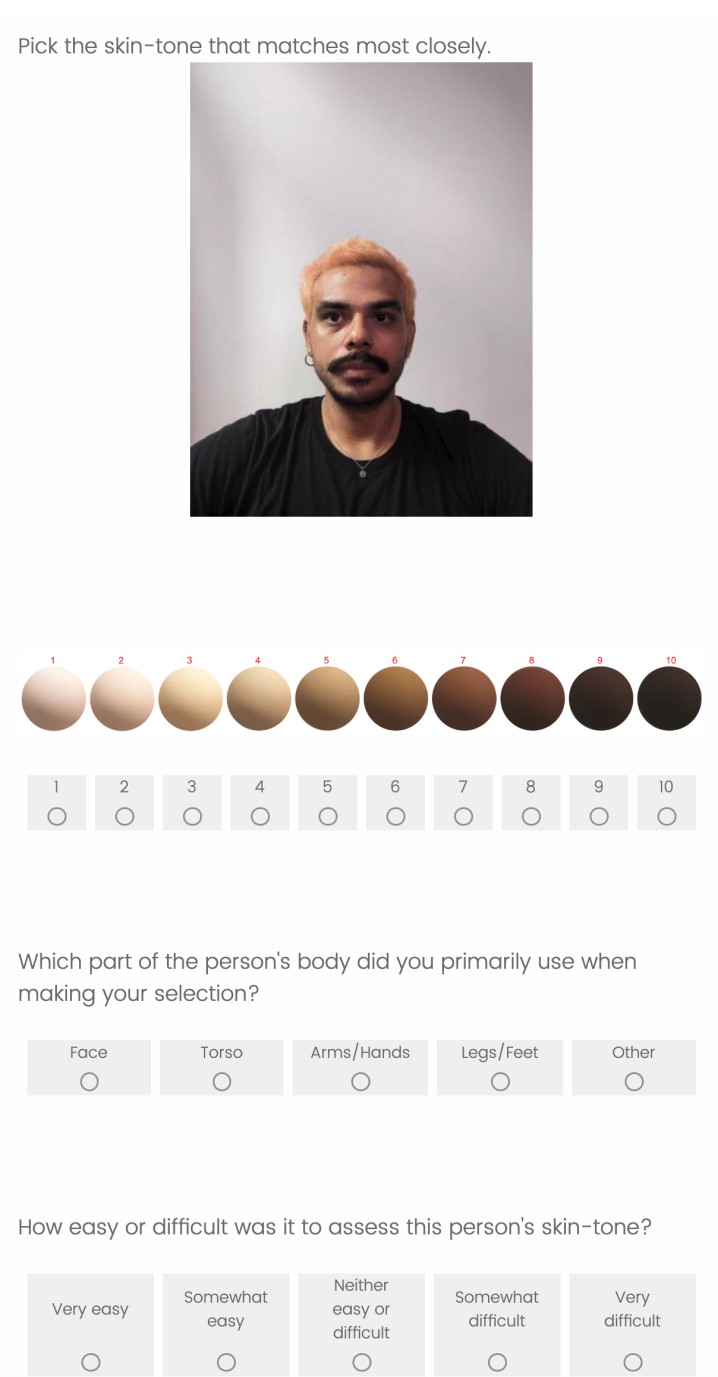

The figure above shows the annotation task as seen by the professional photographers in India and the US. The task was presented in English and all photographers were fluent English speakers. The photographers were sourced through GLG, Gerson Lehrman Group. Each network member of GLG, including the photographers from this study, accept the GLG Terms and Conditions prior to participation.

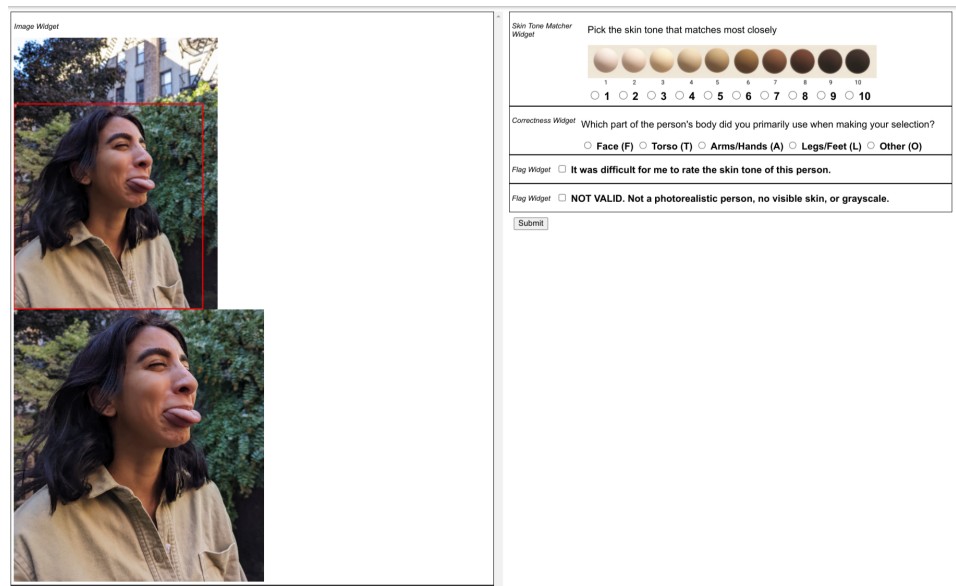

The figure above shows the annotation task as seen by the trained crowdsourced annotators located in India, the Philippines, Brazil, Hungary, or Ghana.

# E   Professional Photographers' Regional Differences

Figure 7 shows the complete Mann Whitney U tests comparing annotations from professional photographers in the US to professional photographers in India. It includes the tests over the *Golden Images* set (Figure 7(a)) and the *Challenging Images* set (Figure 7(b)).

| MST | Subject | Value | $p$ |
|---|---|---|---|
| MST1 | S18 | 7.5 | 0.179712 |
| MST2 | S1 | 19.5 | 0.147499 |
| MST2 | S13 | 23.0 | **0.026888** |
| MST2 | S8 | 18.0 | 0.256839 |
| MST3 | S0 | 18.5 | 0.218758 |
| MST3 | S15 | 20.0 | 0.070701 |
| MST4 | S7 | 12.5 | 1.000000 |
| MST4 | S9 | 21.0 | 0.058758 |
| MST5 | S11 | 14.0 | 0.796253 |
| MST5 | S6 | 11.0 | 0.824848 |
| MST6 | S14 | 24.0 | **0.018119** |
| MST6 | S3 | 8.5 | 0.408308 |
| MST7 | S5 | 9.0 | 0.488422 |
| MST8 | S17 | 9.0 | 0.512412 |
| MST8 | S2 | 6.5 | 0.204024 |
| MST9 | S10 | 12.5 | 1.000000 |
| MST9 | S4 | 6.0 | 0.175151 |
| MST10 | S12 | 9.5 | 0.588809 |

(a) Golden Images

| MST | Subject | Value | $p$ |
|---|---|---|---|
| MST1 | S18 | 13.0 | 1.000000 |
| MST2 | S1 | 18.0 | 0.248213 |
| MST2 | S13 | 13.5 | 0.910670 |
| MST2 | S8 | 14.5 | 0.733730 |
| MST3 | S0 | 20.0 | 0.132622 |
| MST3 | S15 | 9.5 | 0.587594 |
| MST4 | S7 | 19.5 | 0.154424 |
| MST4 | S9 | 16.5 | 0.443194 |
| MST5 | S11 | 20.5 | 0.110492 |
| MST5 | S6 | 13.0 | 1.000000 |
| MST6 | S14 | 22.0 | **0.048632** |
| MST6 | S3 | 6.0 | 0.188667 |
| MST7 | S5 | 13.0 | 1.000000 |
| MST8 | S17 | 4.0 | 0.058758 |
| MST8 | S2 | 6.0 | 0.193059 |
| MST9 | S10 | 10.0 | 0.607236 |
| MST9 | S4 | 8.0 | 0.344659 |
| MST10 | S12 | 11.0 | 0.817361 |

(b) Challenging Images

Figure 7: Mann-Whitney U testing the difference in the annotations of professional photographers in the US versus India. Figure 7(a) tests annotations over the *Golden Images* set. Figure 7(b) tests annotations over the *Challenging Images* set.

## F   Distance from Intended Annotations

Figure 8 shows confusion matrices of Dr. Monk's annotations compared to the median annotations by the crowdsourced annotators.

## G   Pixel-Based Skin Tone

As mentioned in the related work, measuring skin tone from pixel values has limitations (§2). In this experiment we determine the skin tone through a) automated skin tone detection and b) manually cropped images. For each of these methods we look at closest MST skin tone by averaging all skin tone pixels and finding the closest MST point based on the L2 distance in RGB space to the reference colors [14].

### G.1   Automated skin tone detection

Following Cho et al. [14], we mask out non-skin tone pixels using [47]. We find that this method performs poorly as it picks up some background pixels, and does not capture all of the face – particularly dark skin tones (see Figure 9).

Given that this method does not pick out skin tone well, the closest MST values do not accurately represent the subjects skin tone (See Figure 10).

### G.2   Manually cropped images

In addition to the automated skin selection above, we also manually drew boxes around an open portion of skin, and used the pixels from that crop (See Figure 11). As seen in Figure 10, cropped images with only skin tone pixels still do not produce accurate MST annotations.

## H   Maintenance plan

The dataset will be stored on Google Cloud indefinitely. The website skintone.google/mste-dataset will remain updated with the latest version of the dataset. The dataset will be available for download with a valid Google account. Any feedback or found errors in the dataset can be reported through the skintone.google website feedback form. Any errors will be fixed and the dataset will be updated accordingly. Currently we do not plan on releasing a V2.0 version of the dataset.

## I   Author statement and dataset use

Each subject provided consent for their images and videos to be released. TONL has given permission for these images to be released and used for research or annotator-training purposes only (Copyright TONL.CO 2022). The images cannot be used to train machine learning models. Annotations provided with the images can be used for research or annotator-training purposes only. When using the dataset for research, please cite this paper and credit TONL for any images used.

The authors of this paper bear all responsibility in case of any violation of rights during the collection of the data or other work, and will take appropriate action when needed, e.g. to remove data with such issues.

## J   Dataset format

The full dataset can be downloaded from skintone.google/mste-dataset. The download will be a zip folder containing images in `.jpg` format, videos in `.mp4` format, and annotations in a `.csv` format. See the datacard in Appendix C for more information about the fields found in the .csv file.

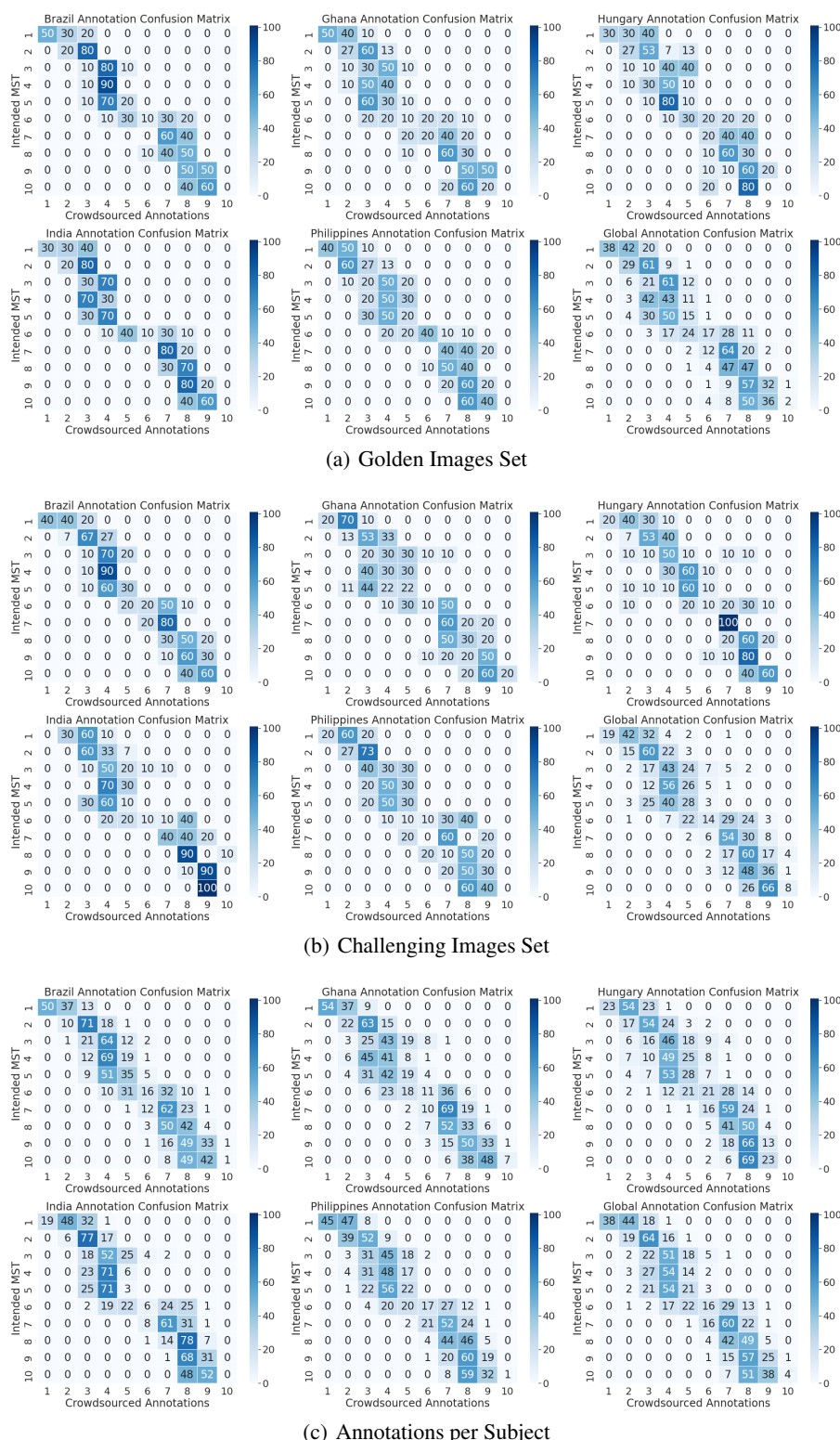

Figure 8: Comparisons between the intended MST value as annotated by the MST creator and the median value selected by crowdsource annotators across subsets of images and different annotator pools.

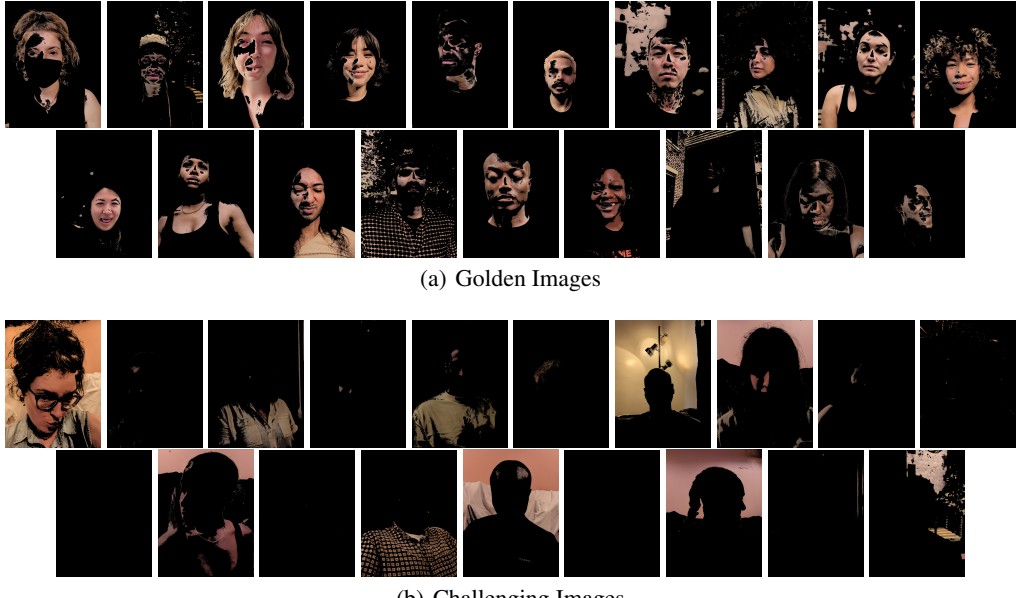

(a) Golden Images

(b) Challenging Images

Figure 9: Detected skin tone following [47]. All pixels that are not skin tone are black.

## K  Annotator compensation

The compensation rates for the expert photographers and the crowdsourced annotators were dictated by the relevant sourcing vendors. Payment rates are determined by fair market value in the different regions. The expert photographers were paid a token honoraria not exceeding $500 by GLG for participation in two 30 minute surveys. Crowdsourced annotators were contracted by the sourcing vendor to work for monthly increments. The rough monthly payment for crowdsourced annotators per region is as follows:

| Location | Per Rater/Mo | Estimated Hourly Wage |
|---|---|---|
| India | ∼$1,100.00 | ∼$6.00 |
| Philippines | ∼$2,000.00 | ∼$11.00 |
| Hungary | ∼$3,500.00 | ∼$19.00 |
| Ghana | ∼$2,300.00 | ∼$13.00 |
| Brazil | ∼$2,300.00 | ∼$13.00 |

Overall, we spent approximately $780k on crowdsourced annotations. The total compensation for the group of expert photographers who provided annotations through GLG was approximately $5000.

## L  Addressing previous reviews

This paper was previously submitted to FAccT 2023 titled, *Consensus and Subjectivity of Skin Tone Annotation for ML Fairness*. The reviewers agreed on the paper's relevance noting the nuance and exploration of skin tone annotation across regions and the implications of the work for practitioners, but also noted some objections and opportunities in their rejection of the submission. We leveraged their feedback for a stronger submission, and below is a summary of their feedback and how the feedback was addressed in this submission.

**Monk Skin Tone Scale**    In the prior submission reviewers raised objections about a missing citation for the Monk Skin Tone scale and who was chosen as a Monk Skin Tone scale expert to provide ground truth annotations. Due to the timing, we were unable to cite the now published Monk Skin Tone scale white paper that describes the scale's development. We have cited it in this paper in reference to the scale.

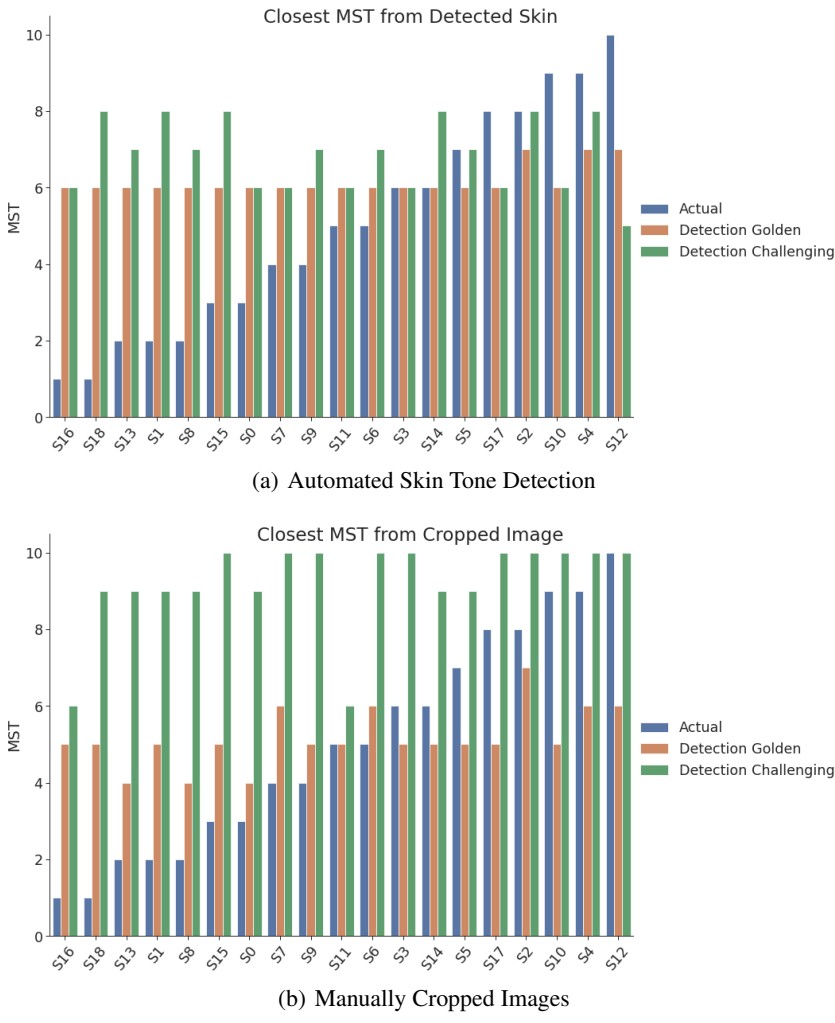

(a) Automated Skin Tone Detection

(b) Manually Cropped Images

Figure 10: Actual MST annotations by the MST creator compared to the closest MST RGB from the average RGB of the detected/selected skin tone pixels.

Relatedly, one reviewer raised a question about the effect of scale length and why a 10-point scale is better or worse than other compositions. This is an interesting question, but out of scope for this paper. In this submission, and the prior one, we focus on people's ability to annotate skin tone on this particular scale due to the recent fairness literature adopting the Monk Skin Tone scale.

**Monk Skin Tone Scale Expert**   In the prior submission reviewers raised objections about the unidentified Monk Skin Tone scale expert, (described as "a Monk Skin Tone scale expert"), and what credentials they possessed to establish ground truth on skin tone, a characteristic we highlight as subjective. In the previous submission we wanted to remain double blind, but in this submission we highlight that Dr. Monk is the Monk Skin Tone scale expert and cite his credentials of academic expertise in skin tone research and the creator of this particular skin tone scale.

**Expert Annotation Exploration**   In the prior submission reviewers raised questions about the selection and limited sample size of experts in the expert annotation portion of the paper. In this submission we clarify why photographers were chosen instead of dermatologists – the type of expert chosen in similar previous research. Photographers tended to assess skin tone in the image while some dermatologists tended to devise the skin tone under clothing which was not depicted in the image (or skin untouched by sun). We also more clearly delineate the expert portion of the paper as exploratory and use the results from this work to establish hypotheses for the properly powered

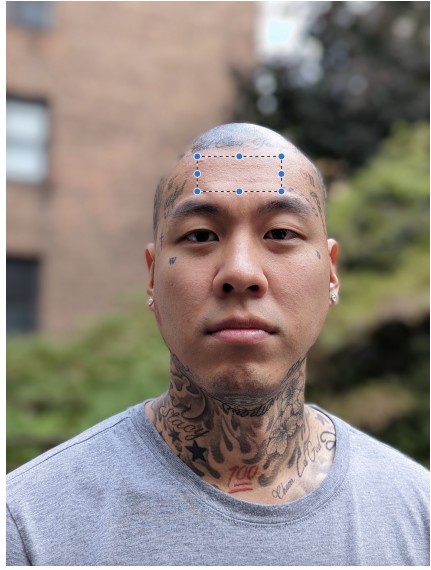
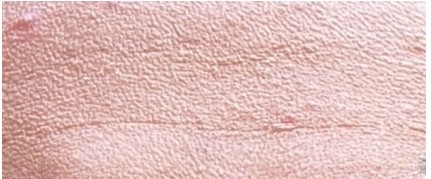

(a) Manually Drawn Box           (b) Resulting Crop

Figure 11: Manually drawn box on an open selection of skin, and the resulting crop of the manual selection.

analyses later in the paper. Finally, we also note an opportunity to more expressly test annotation across various types of experts in the limitations for future consideration.

**Analytic Clarifications**  In the prior submission reviewers sought some additional clarifications in the analyses. At a high-level, we separated the analyses by sample in this version of paper to better delineate the separate analyses, and increase the clarity of the results. We also more clearly define "average median distance" and describe the "1-point discrepancy" threshold citing how they were used in previous research and their purpose in this work. Finally, we made some adjustments to the figures and footnotes where reviewers noted confusion or a preference for more information.

**Clarification of Scope**  In the previous submission we reviewed a broad scope of research including subjective and objective forms of skin tone annotation and in this submission we more clearly focused on subjective annotations to streamline the literature review and highlight the particular focus of this contribution.

**Limitations and Ethical Considerations**  The previous submission did not include an explicit section dedicated to the limitations and ethical considerations of the submission. In this submission we included both to highlight further opportunities to investigate annotation ability quality and the proper usage of the MST-E dataset.

**Monk Skin Tone Scale-Examples Dataset**  One reviewer noted a limitation of the MST-E dataset, it only includes 19 subjects. This is a limitation we accept and acknowledge in the paper, but we are unable to address in this iteration of the dataset. Another reviewer mentioned some additional demographic details like race/ethnicity of the MST-E subjects, but this data was explicitly not collected from the models. Finally, a reviewer mentioned that releasing the annotations would increase the utility of the MST-E dataset for practitioners if we were able to make those publicly available as well. We release the annotations with this paper. You can download the annotations from https://storage.googleapis.com/mste/mste_annotations.csv. See Appendix C for more information.

