# OpenReview forum: "Consensus and Subjectivity of Skin Tone Annotation for ML Fairness"
_NeurIPS.cc/2023/Track/Datasets_and_Benchmarks — NeurIPS 2023 Datasets and Benchmarks Poster_

### Official Review · Reviewer_ph2V · 2023-07-19
**Novel and detailed skin tone scale, and interesting suggestions for skin tone annotation**

**Rating:** 6
**Confidence:** 4
**Correctness:** Yes.
**Clarity:** Yes.

**Strengths:**

- A novel and more detailed skin tone scale (Monk Skin Tone) than Fitzpatrick skin types.
- Interesting suggestions for practitioners in order to scale the data labeling process.

**Additional Feedback:**

Thank you for this great work!

**Documentation:**

Yes.

**Ethics:**

No.

**Limitations:**

- Dr Monk is the inventor of the Monk Skin Tone scale. However, I believe that Dr Monk being the sole annotator for ground truth is a significant limitation. In most instances, a robust dataset is annotated by numerous domain experts. I always have concerns when a dataset is labeled by a single individual. I would love to learn more about Dr Monk's annotation process, how long did it take him to annotate these photos, what consistent was Dr Monk at this task, etc.
- I agree that photographers are very sensitive to skin tones. However, I believe that it would have strengthened the study if the authors included dermatologists in the annotation experiment.
- The dataset is based on 18 subjects with memorable facial features. It is possible that annotation experts remembered their rating for each subject and tried to be consistent. More subjects and less photos per subject might have been a better study design.

**Opportunities For Improvement:**

- I would love to learn more about technical acquisition details and (if any) preprocessing procedures of images in this dataset.
- The authors showed that annotators from different geographical regions will annotate skin tones slightly differently. I wonder either this difference can be solely attributed to regional cultural contexts, and either the annotators' own skin tone comes into play. Did the authors look at this specific issue? Did the US experts come from a diverse cultural background?
- It would have been interesting for the authors to show a few applications of this dataset. Currently, I have difficulty seeing either this dataset will be used for ML training or benchmark.
- In page 7, the authors wrote: "Using the global pool, the consensus annotation agreed within 1 point for all but one subject (94.7%)." Which subject was this? And is there an explanation behind this?

**Relation To Prior Work:**

Yes.

**Summary And Contributions:**

I would like to thank the authors for this novel and detailed skin tone scale (Monk Skin Tone), and interesting suggestions for practitioners in order to scale the data labeling process.

---

> ### Author Response · Authors · 2023-08-18
> **Response to reviewer**
>
> Thank you so much for your very detailed review! Please take a moment to look at the global answer which addresses some of your questions and concerns. In addition to those answers we address your other questions or concerns here:
> 1. On annotator demographics
>     - We did not collect demographic data from our annotators to reduce the amount of sensitive data in this study. However, we did request a diverse range of photography experts from GLG and can confirm based on their public profiles they are a demographically diverse group of photographers.
> 2. “Using the global pool, the consensus annotation agreed within 1 point for all but one subject (94.7%).” Which subject was this? And is there an explanation behind this?
>     - This subject was Subject 12. Subject 12's golden annotation is MST10. Coming to a consensus on the end points of the scale is a difficult task due to a number of reasons. The first of which is mathematical, median works less well at end points of a scale. The other factors could be human based. We have already seen that different regions annotate skin tone differently. Darker skin tones in particular have a very complex history which may make annotators less willing to assign the darkest MST value to a person in an image.
> 3. The dataset is based on 18 subjects with memorable facial features. It is possible that annotation experts remembered their rating for each subject and tried to be consistent
>     - While it is possible that raters remembered their rating for each subject - we believe it is unlikely. These images were mixed in with a large number of other images over a number of days (approximately 1.5 weeks). Raters on average take ~30 seconds to answer a skin tone question. Given the large pool of raters, the large number of images mixed in with the MST-E dataset, the number of days this task was done over, and the time taken per question, the risk of memorization of specific annotations for individuals is greatly reduced.

---

> > ### Comment · Area_Chair_zc4i · 2023-08-29
> > **Response to author comments and confirming rating for submission 841 / Consensus and Subjectivity of Skin Tone Annotation for ML Fairness**
> >
> > Dear reviewer ph2V,
> >
> > If you have a moment, I would greatly appreciate if you could briefly respond to the author comments on your review, and (if desired) update your rating accordingly?
> >
> > Thank you so much!
> >
> > Kindest regards,
> > Jochen Weber (Area Chair zc4i)

---

> > ### Comment · Reviewer_ph2V · 2023-08-29
> >
> > Thank you so much for the clarification! Looking forward to learn more about research on skin tone!

---

### Official Review · Reviewer_jgJ9 · 2023-07-21
**Consensus and Subjectivity of Skin Tone Annotation for ML Fairness**

**Rating:** 6
**Confidence:** 4
**Correctness:** It seems correct to me.
**Clarity:** The paper is sufficiently well-written.

**Strengths:**

Comprehensive experiments on annotations provided by annotators from different geographical regions.
The analysis of annotations of skin tone by annotators across conditions.



**Additional Feedback:**

N/A

**Documentation:**

The paper provides MST-E dataset.

**Limitations:**

The lack of an explanation on how this work contributes to the advancement of the machine learning and the absence of description and experiments regarding its application in machine learning tasks are evident.

**Opportunities For Improvement:**


Adding some insightful descriptions or experiments about how this work can assist machine learning in different scenarios or tasks and showing comparison between this work and existing methods.

**Relation To Prior Work:**

The authors clearly discuss how this work differs from previous papers in the Background and Related Work.
It appears to be a novel work.

**Summary And Contributions:**

This paper shows how to annotate skin tone reliably on the MST scale by both trained crowdsourced annotators and subject-matter experts. Furthermore, it provides suggestions for practitioners who are pursuing a task of MST scale annotation and releases the MST-E dataset which contains MST annotations provided by the creator of the MST scale to support further research.

---

> ### Author Response · Authors · 2023-08-18
> **Response to reviewer**
>
> Thank you so much for your review! If you could take a moment to review the global response as it provides a response to your main concern.

---

> > ### Comment · Reviewer_jgJ9 · 2023-08-29
> >
> > Thx for your reply. My concerns have been solved. Good luck!

---

> > > ### Comment · Area_Chair_zc4i · 2023-08-29
> > > **Final rating updated for submission 841 / Consensus and Subjectivity of Skin Tone Annotation for ML Fairness**
> > >
> > > Dear reviewer jgJ9,
> > >
> > > If you have a moment, can you please confirm (and ensure that you have updated) the rating for this submission? Thank you so much! :)
> > >
> > > Best regards,
> > > Jochen Weber (Area Chair zc4i)

---

### Official Review · Reviewer_JyxF · 2023-07-21
**Practical and well explored recommendations for perceived skin tone annotations**

**Rating:** 7
**Confidence:** 5

**Strengths:**

The authors provide a very thorough and convincing analysis as to variation in perceived skin tone annotations. The MST-E dataset will be useful for researchers hoping to gather fairness related annotations for images and videos. The ground truth annotations created by Dr. Monk will create an excellent baseline to train annotators against, and could lead to more standardized annotations across datasets.
The paper offers and concrete and practical suggestions for researchers hoping to gather perceived skin tone annotations, and provides experimental evidence for their suggestions.

**Additional Feedback:**

N/A

**Clarity:**

The paper is well written and easy to follow. The experimental design and analysis is easy to understand.

**Correctness:**

The authors thoughtfully included skin tone annotation comparisons from three groups: 1) the creator of the monk skin tone scale 2) photography experts 3) a pool of non-expert annotators. Their dataset is well designed to cover a series of lighting and angle conditions. The authors took procedures to control the lighting conditions and skin tone as much as possible - using photos only from one day within a four hour period of time.

**Documentation:**

A dataset card including details about the availability, license, and maintenance plan is included in the appendix.

**Ethics:**

No.

**Limitations:**

The authors note that the dataset is limited in size. The authors note the potential for misuse of skin tone annotations, and limit the use cases of the dataset to prevent misuse.

While the work is quite interesting, the range of impact is rather niche: it most directly lends it self to researchers hoping to build fairness vision datasets using annotators.

**Opportunities For Improvement:**

The authors have collected a dataset of images of people and their Monk Skin tone scale in a variety of conditions include challenging lighting conditions. The authors note that they have highly accurate annotations from the creator of the scale for these images. The authors specifically prohibit training on the images, as is common with other fairness related perceived skin tone annotations [1]. However, the authors do not mention any other allowed or prohibited sue cases out side of allowing training of annotators and prohibiting model training. Do the authors allow use of the images and their associated annotations for evaluation of computer vision models? While the dataset is small in size, the accuracy of the labels could make it a useful benchmark for researchers.

The authors suggest gathering annotations from 10 annotators per image, which is significantly more than three annotations, which is the current common practice. This would make annotating a dataset of a fixed size significantly more expensive, and may not be practical for groups. It would be useful if the authors could give additional recommendations as to how best to counteract annotator bias with limited resources.

[1] Schumann, Candice, et al. "A step toward more inclusive people annotations for fairness." Proceedings of the 2021 AAAI/ACM Conference on AI, Ethics, and Society. 2021.

**Relation To Prior Work:**

While apparent skin tone annotations have been collected for vision datasets in the past, this work studies the process of collecting the annotations instead the analysis of models with the annotations.  This work validates the need for a geographically diverse annotator pool.

**Summary And Contributions:**

The paper represents a thorough and thoughtful analysis of annotating the perceived skin tone of people within in an image. This is a task commonly done when researchers are creating computer vision fairness annotations/datasets. The paper represents a clear and thorough analysis of and its main contributions are:

1) Findings that lead to practical suggestions for researchers: It is sufficient to use annotators for annotating skin tone opposed to experts. It is important to source geographically diverse annotators to compensate for annotator bias. The authors suggest collecting annotations from 10 annotators.

2) Release of a golden set of images of people and their perceived skin tone on the Monk skin tone scale, that can be used by other researchers to train their annotators to annotate perceived skin tone.

---

> ### Author Response · Authors · 2023-08-18
> **Response to Reviewer**
>
> Thank you so much for your very detailed review! Please take a moment to review the global response to all reviewers as it addresses some of your questions and concerns. In addition we provide more specific comments here:
> 1. 10 annotators per image - It would be useful if the authors could give additional recommendations as to how best to counteract annotator bias with limited resources.
>     - If the budget is limited, we would suggest filtering for images with good lighting conditions (or even images with ok lighting conditions). As mentioned in Section 6 (under replications), we find that images with good lighting conditions can be annotated with just 5 annotators. We will be sure to make this more clear in the revision of the paper.
>     - A potential alternative for very restricted budgets would be to use a large and geographically diverse pool of annotators but have only a subset of annotators provide explicit judgements for each image. One would then impute missing annotations based on a machine learned model personalized for each annotator. Similar approaches have been tested for other types of subjective annotations (in NLP, cite A; for face similarity, cite B) but not yet for skin tone. Because this adds design of an ML system, we consider it out of scope for this paper but will add it to our discussion of future work.
>
> References:
>
> A)  Aida Mostafazadeh Davani, Mark Díaz, Vinodkumar Prabhakaran. "Dealing with Disagreements: Looking Beyond the Majority Vote in Subjective Annotations." Transactions of the Association for Computational Linguistics, 2022.
>
> B)  Andrews, Jerone TA, Przemyslaw Joniak, and Alice Xiang. "A View From Somewhere: Human-Centric Face Representations." ICLR, 2023.

---

### Official Review · Reviewer_8mCy · 2023-07-28
**The paper proposes a very interesting study about the subjectivity of skin tone annotations. I liked the paper much, but I think this work is not as mature as to be published in this track mainly for two reasons: a limited number of human subjects; and a missing computer vision application that shows the potential of this dataset for a pattern recognition task.**

**Rating:** 6
**Confidence:** 3
**Clarity:** The paper overall is well written.

**Strengths:**

- The presented dataset has been collected through a rigorous procedure. It has been validated also thanks to different group of experts.
- It is a relevant paper for practitioners
- it is very rigorous way for approaching the subjectivity of skin tone annotation.
- The dataset is publicly available for further research.
- The association of MSE-E to the MSE scale is a very good idea since it maps the annotations to a formal, and well known, scale.

**Additional Feedback:**

See comments above

**Correctness:**

The proposed dataset is well constructed and procedure for labeling is well described.

**Documentation:**

The dataset is available for download

**Ethics:**

Should be checked carefully especially regarding the right/proper use of the dataset

**Limitations:**

The proposed dataset seems interesting, however there are many critical aspects, such as:
- Missing Computer Vision perspective
- Limited number of subjects;
- Missing discussion about the possibility of increasing the number of subjects.

**Opportunities For Improvement:**

- The focus of the abstract is a bit misleading with respect to the content of the paper. It is not clear why the authors do not mention they propose a new dataset in the abstract.
- The proposed methodology and dataset is very interesting. However, the proposed dataset includes a limited number of subjects so highlighting a potential problem of under-representation of the reality. This is a major concern.
- Authors discuss about ML fairness in Computer Vision (CV), however they do not present any application of CV methods on the dataset to show the importance of this dataset for the CV community.
- The confusion matrices (supplemental material Figure 7) show Dr. Monk’s annotations compared to the median annotations by the crowdsourced annotators. The figure show that in some cases the consensus is not that high. Anyway, this figure should be commented in the main paper.

**Relation To Prior Work:**

The paper is well grounded on previous work.

**Summary And Contributions:**

The paper discusses recent advancements in computer vision fairness, focusing on datasets that incorporate perceived attribute signals like gender presentation, skin tone, and age. These attributes are crucial for various tasks but are often challenging to annotate, particularly skin tone. The perception of skin tone can be influenced by technical factors (such as lighting conditions) and social factors related to the annotator's personal experiences.

To study the subjectivity of skin tone annotation, the researchers conducted a series of experiments using the Monk Skin Tone (MST) scale. They involved a small group of professional photographers and a much larger pool of trained crowdsourced annotators. The results showed that annotators can consistently and reliably annotate skin tone, aligned with an expert using the MST scale, even under challenging environmental conditions.

The study also revealed that annotators from different geographic regions tend to apply different mental models of MST categories, leading to systematic variations in annotations across regions. In light of this finding, the paper suggests that practitioners should use a diverse set of annotators and increase the replication count for each image when annotating skin tone for fairness research. This approach helps to mitigate potential biases and improve the overall reliability of the skin tone annotations in computer vision applications.

---

> ### Author Response · Authors · 2023-08-18
> **Response to Reviewer**
>
> Thank you so much for your very detailed review! Please take a moment to review the global response to all reviewers as it addresses some of your questions and concerns. In addition we provide more specific comments here:
> 1. Limited number of subjects
>     - We agree that this limited number of subjects is definitely a limitation of the dataset. However, when comparing the distribution of skin tones of the MST-E dataset to an in-the-wild dataset such as open images, the MST-E dataset is much more uniform across skin tone when compared to open images which is heavily over saturated with lighter skin tones - specifically skin tone 2. We will add in a plot of this in the appendix.
>     - Extending this dataset to more subjects is a part of future work.
>
> We have an additional question about your selection of “significant ethics concerns”. We have been extremely careful with the collection of this dataset. All subjects have consented to their images being used for research (including evaluations) and for training human annotators on skin tone annotations. In addition, we have consent from the annotators to release the annotations. We have a terms of use for this dataset - that being that the images cannot be used to train machine learning models. That being said, as with any dataset release on the internet, we cannot fully account for bad actors. We do have a form on the website for people to ask questions or report any concerns. This process is detailed in the maintenance plan in the appendix as well as in the data card in the appendix. Could you please expand on your ethics concern? We would like to proactively address these.

---

> > ### Comment · Area_Chair_zc4i · 2023-08-29
> > **Confirming the rating for submission 841 / Consensus and Subjectivity of Skin Tone Annotation for ML Fairness**
> >
> > Dear reviewer 8mCy,
> >
> > Can you please take a moment and confirm (and, if desired, update) the rating for this submission (after review)? Thank you so much!
> >
> > Best regards,
> > Jochen Weber (Area Chair zc4i)

---

> > ### Comment · Reviewer_8mCy · 2023-08-29
> > **Response to the Authors' comments**
> >
> > Thank you for your detailed reply. I also appreciated your idea to update the abstract (as you mentioned in the global comment above).
> > Regarding the ethics concerns, these mainly regard a potential bad use of the images you provide. For instance the creation of new synthetic images by using Generative AI, or synthetic videos of the subjects of your dataset that perform different scenes.
> > I still have a concern, partially addressed  in the global comment, regarding the use of this dataset in the CV community. Can you add potential uses you mentioned in the paper?

---

> > > ### Author Response · Authors · 2023-08-31
> > > **Response to Reviewer**
> > >
> > > Thank you for the reply! Yes, we have uploaded a revision to the paper with all updated sections highlighted.

---

### Author Response · Authors · 2023-08-18
**Global Rebuttal**

Firstly, we would like to thank all of the reviewers and the area chair for their in-depth review of this paper! We really appreciate all of the feedback! We will address some common questions/comments here for all reviewers and will then reply with specific comments to individual reviewers.

---

> ### Author Response · Authors · 2023-08-18
> **On dataset use and relevancy to ML and CV**
>
> - First and foremost this paper and dataset hopes to provide human annotator training material and best practices for annotating skin tone. As mentioned by Reviewer JyxF21, annotating skin tone is a common task when researchers are creating computer vision fairness annotations/datasets. Having a consistent way in which to collect annotations across datasets helps create consistency across other datasets and benchmarks, hence the reason for submitting to this particular track. At the moment there have been two major uses of Monk Skin Tone in research papers. The first of which is in the casual conversations dataset which does not mention how MST annotations were collected (cite A). The second of which is DALL-E evaluations (cite B) which annotates MST using an automated method we do not suggest based on our research findings (see appendix G). Our research and methods would help improve annotations and downstream research in both of these cases. MST is a recently announced alternative to the coarser skin tone scales used in prior computer vision datasets. We anticipate future work will also incorporate MST annotations following the lead of these two early adopters, making this a key moment for setting the direction of the field. In addition the call for papers in this track specifically call for "Data-centric AI methods and tools, e.g. to measure and improve data quality or utility, or studies in data-centric AI that bring important new insight." and "Frameworks for responsible dataset development, audits of existing datasets, identifying significant problems with existing datasets and their use". We believe that this paper falls into a combination of both of these asks.
> - There has been some confusion regarding how this dataset can be used. Although this dataset __cannot__ be used to train machine learning models, and due to the size would not be useful for training tasks, it can be used for two important tasks: training human raters on MST skin tone annotating, and performing disaggregated fairness evaluations on models for both MST as well as lighting conditions. We will update both the description of the dataset in the main paper as well as the datacard found in the appendix to reflect this more clearly.
> - Disaggregated fairness analysis is becoming far more prevalent in both industry and academia and, as mentioned by Reviewer JyxF21, annotating skin tone is a common task when researchers are creating computer vision fairness annotations/datasets (cite A, B, C). The focus on responsible model creation from data curation to evaluations have become evident in regulation and policy discussions (cite D, E, F), thus, understanding different human attributes and how they affect or are affected by large models may become a standard need for all model creation and usage. In computer vision specifically, we foresee skin tone becoming a significant part of this conversation.
>
> References:
>
> A) Porgali, Bilal, et al. "The Casual Conversations v2 Dataset." Proceedings of the IEEE/CVF Conference on Computer Vision and Pattern Recognition. 2023.
>
> B) Cho, Jaemin, Abhay Zala, and Mohit Bansal. "DALL-EVAL: Probing the Reasoning Skills and Social Biases of Text-to-Image Generative Models." arXiv preprint arXiv:2202.04053 (2022).
>
> C) Buolamwini, Joy, and Timnit Gebru. "Gender shades: Intersectional accuracy disparities in commercial gender classification." Conference on fairness, accountability and transparency. PMLR, 2018.
>
> D) Regulation (EU) 2016/679 of the European Parliament and of the Council of 27 April 2016 on the protection of natural persons with regard to the processing of personal data and on the free movement of such data, and repealing Directive 95/46/EC (General Data Protection Regulation) [2016] OJ L 119/1
>
> E) “MEPs Ready to Negotiate First-Ever Rules for Safe and Transparent AI: News: European Parliament.” MEPs Ready to Negotiate First-Ever Rules for Safe and Transparent AI, European Parliament, 14 June 2023, www.europarl.europa.eu/news/en/press-room/20230609IPR96212/meps-ready-to-negotiate-first-ever-rules-for-safe-and-transparent-ai.
>
> F) “Blueprint for an AI Bill of Rights.” The White House, The United States Government, 16 Mar. 2023, www.whitehouse.gov/ostp/ai-bill-of-rights/.

---

> ### Author Response · Authors · 2023-08-18
> **On expert annotators (Dr. Monk + Dermatologists)**
>
> - In this paper we used Dr. Monk’s annotation as our golden annotations, but it is important to clarify that Dr. Monk was not the sole annotator. Rather, one of the authors, Auriel Wright, supervised selection of the models for the dataset based on skin tone and Dr. Monk’s annotations served as a verification of the skin tone assessment. In addition, we are releasing all of the annotations and not just Dr. Monk's, so each image will have the golden annotation used in the paper and the range of annotations provided by professional annotators. We will make sure to clarify this in the revision of the paper.
>  - In an earlier pilot we asked various types of skin tone experts including dermatologist and photographers to complete a similar annotation task (as detailed in the first paragraph of Section 5). We found photographers would approach the task as described and select the person’s skin tone in the image, doing some correction for adverse or unnatural light. We found that some dermatologists would attempt to select the target’s skin tone as it would appear beneath clothing - how skin tone assessment is done in a clinical setting. Given the goal of assessing skin tone in imagery - the perception of a person’s skin tone in the image, and avoiding a potential confound on annotation style, we focused on exploring the annotations among photographers in this paper.
> - However, we agree with Reviewer # 1, it would have been an interesting experimental condition to compare the skin tone annotations of various types of experts. Based on the initial pilot, we hypothesize that different types of experts may a) approach the task differently based on their discipline and b) this might result in differences in annotation. In this paper we wanted to focus on an exploration of regional differences within a single discipline and not differences between disciplines – as to mirror differences we hypothesized to find different professional annotators. We think a future research focusing on annotation across various disciplines would be a great gain to the literature - especially one that could tackle both differences in region and discipline concurrently.

---

> ### Author Response · Authors · 2023-08-18
> **On Abstract**
>
> Almost all of you have suggested updating the abstract. We will update the abstract to include the positioning of this paper (as described above) as well as include information about the dataset. Stay tuned for a revised paper upload.

---

### Decision · Program_Chairs · 2023-09-22

**Decision:**

Accept (Poster)

**Comment:**

Submission: Skin Tone Annotation Consensus (**#841**) Bottom line: accept (as poster)

**Summary**: Skin tone annotation is an incredibly challenging -- and increasingly important -- aspect of doing skin (disease related) research across populations of different skin types and tones. The current (often employed) standard is and remains the Fitzpatrick skin types/scale methodology, which however is flawed. The authors provide strong evidence for the validity of a novel and yet simple (10-category nominal) system, the Monk scale -- with Dr. Monk included on the list of authors, providing additional support for this publication. While several minor concerns remain, the AC believes that offering this annotation study and benchmarking methodology will drive the necessary improvements on skin type research forward significantly, and thus urges the SAC/PCs to accept the manuscript for (poster) publication.

Pros: two separate "user" groups (skin image experts in form of photographers with often greater care for color as an ontological/perceptual category, and novices) were asked to annotate the dataset of images, which comprise a specially constructed gold standard set, as well as a convenience sample of images of skin. Together with the analyses provided, the improvement on the original methodology (Monk scale, in collaboration with Dr. Monk) by creating patches showing the color under different lighting conditions, and the rigorous collection procedure, this dataset provides very good evidence and instructions for further research, creating a solid basis for future studies.

Cons: the dataset may be very niche (audience from the NeurIPS community is the relatively small overlap of automatic image processing *and* fairness *and* skin color as a hard-to-assess percept/concept), and from the title, the general difficulty that skin tone offers in dermatological research isn't as clear; **this paper might be better placed in an outlet that targets dermatologists with a strong research interest; so long as the SAC/PCs do not consider this to be a limiting factor, the recommendation to accept as a poster stands.**

Decision: given that this topic (skin tone, and lack of clear direction in the dermatology research domain with addressing the dearth of skin tone/color research) is of great interest and import **in the dermatology community**, the AC believes that publishing is warranted, albeit maybe better in a different outlet (this is difficult for me to assess, w.r.t. how much NeurIPS wishes to support smaller niche communities...). That notwithstanding, the manuscript and dataset as such **are important contributions**.

----

Reviewer 8mCy: the major concerns are poor representation of the subject to the overall computer vision (CV) audience; some of this has been addressed (by rewrites), and the reviewer seems largely satisfied; in addition the reviewer raised ethics concerns (consent of participants), which is of note; the AC believes the authors have covered this as best they could.

----

Reviewer JyxF: major concerns are around use cases, dataset size, and 10-annotator recommendations; in my reading of the rebuttals and the reviewed manuscript, these points are satisfactorily addressed (although the reviewer did not respond to the authors' rebuttal, and so I cannot say whether the reviewer would share this assessment).

----

Reviewer jgJ9: major concern is around a lack of clear descriptions of use cases for the machine learning (or CV) communities; I agree with the reviewer, and while the authors addressed this, the issue of "best outlet" for this publication remains (see above); the AC believes in the high value of the work, and supports publication, just unsure of outlet...

----

Reviewer ph2V: major concern is that Dr. Monk as the sole "ground truth" annotator is problematic; importantly, these ground truth annotations are **not** the major point of the publication (but the methodology and benchmarking method); as such the AC does not believe this to be grounds for rejecting the publication.

----

AC concerns:  the authors have addressed all concerns to my satisfaction (some concerns are limitations, which are equally addressed to my satisfaction).